

# A conceptual prediction model of seasonal drought processes using atmospheric and oceanic Standardized Anomalies: application in four recent severe drought events in China

Zhenchen Liu[1], Guihua Lu[1], Hai He[1], Zhiyong Wu[1], Jian He[2]

[1] College of Hydrology and Water Resources, Hohai University, Nanjing, China.
[2] Hydrology and Water Resources Investigation Bureau of Jiangsu Province, Nanjing, China.

*Correspondence to*: Hai He (hehai_hhu@hhu.edu.cn)

**Abstract.** Reliable drought prediction is fundamental for end water managers to develop and implement drought mitigation measures. Considering the idea that drought development is closely related to the spatial-temporal evolution of large-scale circulation patterns, we develop a conceptual prediction model of seasonal drought processes based on atmospheric/oceanic Standardized Anomalies (SA). Empirical Orthogonal Function (EOF) analysis was firstly applied to drought-related SA of 200 hPa/500 hPa geo-potential height (HGT) and sea surface temperature (SST), respectively. Subsequently, SA-based predictors were built based on the spatial configuration of the first EOF modes. This drought prediction model is essentially the synchronous statistical relationship between 90-day-accumulated atmospheric/oceanic SA-based predictors and 3-month SPI (SPI3), calibrated by the simple method of stepwise regression. It is forced by seasonal climate forecast models like the NCEP Climate Forecast System Version 2 (CFSv2). It can make seamless drought prediction for operational use after being calibrated year-by-year. Model application during four recent severe drought events in China indicates its good performance at predicting seasonal drought development, despite its weakness in predicting drought severity. Therefore, it can provide some valuable information and is a worthy reference for seasonal water resource management.

## 1 Introduction

Drought is an economically and ecologically disruptive natural hazard that profoundly impacts water resources, agriculture, ecosystems, and basic human welfare (Dai, 2011). In recent years, extreme drought events have had terrible effects worldwide. The 2011 East Africa drought led to famine and severe food crises in several countries, affecting over nine million people (Funk, 2011). As part of the 2011–14 California Drought, the drought in 2014 alone cost California $2.2 billion in damages and 17000 agricultural jobs (Howitt et al., 2014). China has also suffered from extreme drought events, such as the 2009/2010 severe drought in southwest China (Yang et al., 2012), the 2011 spring drought in the Yangtze River basin (Lu et al., 2014), and the 2014 summer drought in North China (Wang and He, 2015). Because drought is a costly and disruptive natural hazard,





reliable drought prediction is fundamental for end water managers to develop and implement feasible drought mitigation

measures.

Drought is generally predicted using two types of methods: model-based dynamical forecasting and statistical prediction. Dynamical forecasting mainly relies on computed corresponding drought indicators like the Standard Precipitation Index (SPI; McKee and Kleist, 1993), based on forecasted precipitation retrieved from seasonal climate forecast models (Yoon et al., 2012;Mo and Lyon, 2015;Dutra et al., 2013;Dutra et al., 2014). Although dynamically predicted precipitation is useful

information about drought situation, especially for short-term forecasting, it also contains high levels of uncertainty and limited skill with respect to long lead times (Wood et al., 2015;Yoon et al., 2012;Yuan et al., 2013). Statistical drought prediction, on the other hand, can be seen as an additional source of prospective drought information (Behrangi et al., 2015;Hao et al., 2014). Different from the physically complex processes of coupled atmosphere-ocean models used for dynamical prediction, statistical drought prediction models are relatively simple but also perform well. They consist of input variables, methodology,

and prediction targets (Mishra and Singh, 2011).

Reasons for good and effective performance of statistical models include methodology improvements and drought-related climate indices used as input variables. To date, much attention has been paid to methodology improvements. Taking advantage of probabilistic and temporal-evolution features of input variables, statistical drought prediction models are mainly forced by probability or machine-learning methods, such as the ensemble streamflow prediction (ESP) method (AghaKouchak, 2014),

Markov Chain- and Bayesian Network-Based Models (Aviles et al., 2015;Aviles et al., 2016;Shin et al., 2016), neural network, and support vector models (Belayneh et al., 2014). In addition to method improvement, climate indices act as representatives of large-scale atmospheric or oceanic drivers of precipitation, partly responsible for effective model performance. These climate indices include typical atmospheric and oceanic circulation patterns, such as the North Atlantic Oscillation (NAO; Hurrell, 1995) and El Niño-Southern Oscillation (ENSO; Ropelewski and Halpert, 1987), which have been widely used for

drought prediction in different seasons and regions (Behrangi et al., 2015;Bonaccorso et al., 2015;Chen et al., 2013;Mehr et al., 2014;Moreira et al., 2016).

These inherent climate indices like the NAO index and the NINO 3.4 index are simple, explicit, and widely used, leading them to be the primary indices used for drought prediction. Additionally, based on the relationship between drought indices and potential atmospheric or oceanic circulation patterns, some researches have also discovered large-scale circulation patterns

that are closely related to regional droughts or have structured new drought predictors (Funk et al., 2014;Kingston et al., 2015). For instance, after the discovery of the two dominant modes of the East African boreal spring rainfall variability that are tied to SST fluctuations, Funk et al. (2014) further determined that the first- and second-mode SST correlation structures were related to two SST indices that could be used to predict East African spring droughts.

Similarly, potential atmospheric and oceanic circulation patterns, which are closely related to regional droughts, are also used

to construct drought predictors in the present study. Considering that the development of drought processes is closely related to the spatio-temporal evolution of large-scale circulation patterns, we constructed predictors based on anomalous spatial configurations. Because precipitation-inducing circulation patterns usually occur in the troposphere, predictors can be built



based on sea surface temperature (SST) and 200 hPa/500 hPa geopotential height (HGT), reflecting information from different levels of the troposphere. Subsequently, all predictors during different drought processes and 3-month SPI (hereafter SPI3)

were used for calibration of the synchronous stepwise-regression relationship. The model can be forced by dynamically predicted SST and 200 hPa/500 hPa HGT conditions, indicating that its lead time depends on that of climate prediction models. Based on angle comparison of predicted prospective SPI3 curves, we developed rules of drought outlook.

Overall, the objectives of this study were to (1) use SPI3, to capture severe and extreme drought processes; (2) conduct Empirical Orthogonal Function (EOF) analysis on SA of drought-related 200 hPa/500 hPa HGT and SST and then structure

SA-based predictors; (3) build the synchronous stepwise regression relationship between 90-day-accumulated SA-based predictors and SPI3; (4) propose an objective method of drought outlook based on angle comparison of predicted prospective 90 day SPI3 curves; and (5) simulate and predict four severe seasonal drought processes in China, using the National Centers for Environmental Prediction / National Center for Atmospheric Research (NCEP/NCAR) Reanalysis datasets and the NCEP Climate Forecast System Version 2 (CFSv2) operationally forecasted datasets, to investigate the perfomance of the proposed

model.

## 2 Data

The precipitation data used is the second-version Dataset of Observed Daily Precipitation Amounts at each $0.5°\times0.5°$grid point in China during 1961−2014 (http://data.cma.cn/data/detail/dataCode/SURF_CLI_CHN_PRE_DAY_GRID_0.5.html ), which was kindly provided by the Climate Data Center (CDC) of the National Meteorological Information Center, China

Meteorological Administration (CMA). It was initially used to calculate area-averaged precipitation and SPI3 in North China, East China and Southwest China (Fig. 1), which are three Chinese drought study regions to investigate in this study. Atmospheric anomalies were diagnosed with respect to the NCEP/NCAR Reanalysis datasets, which has a resolution of $2.5°\times2.5°$at 17 pressure levels, extending from January 1948 to present (Kalnay et al., 1996). The National Oceanic and Atmospheric Administration (NOAA) High Resolution SST dataset, which has a spatial resolution of $0.25°\times0.25°$and

extends from September 1981 to present (Reynolds et al., 2007), were used for SST anomaly analysis. Additionally, the NCEP Climate Forecast System Version 2 (CFSv2; Saha et al., 2014) was introduced to verify operational performance of the conceptual model proposed. Since CFSv2 began on 1 April 2011, some drought events that occur before this date were forced with the CFS reforecast output. All the reforecast and forecasted datasets are accessible on the website (https://nomads.ncdc.noaa.gov/modeldata/).





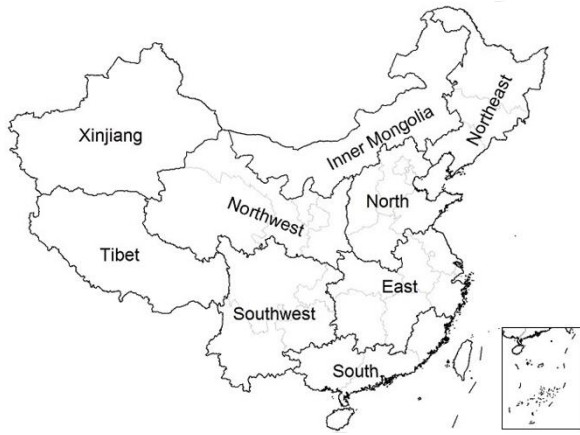


**Figure 1.** The geographical distribution of China's nine drought study regions (black solid curves) and provinces (light grey curves).

## 3 Method

### 3.1 Use SPI3 to capture severe and extreme drought processes

SPI3 was used as the drought index for seasonal drought recognition and prediction in this study. Traditionally, 3-month SPI
is computed based on monthly precipitation aggregated at the 3 month scale. However, to obtain precise start and end dates of
drought processes, we chose an acceptable method recommended by the World Metrological Organization (2012), in which
SPI calculation is based on 3 month moving window (90 day in practice) of area-averaged precipitation data and is updated
everyday. For instance, SPI3 on 1 April 1999 was calculated using the cumulative area-averaged precipitation amount from 2
January 1999 to 1 April 1999. The period for SPI3 calculation is 1979–2014.
Similar to the rules of SPI grade division recommended by the World Metrological Organization (2012), rules in our study are
shown in Table 1. Drought processes were identified when the SPI3 values were below -0.50 for more than 30 consecutive
days. Each daily value of the recognized drought process was assigned to the corresponding SPI3 grade (e.g., severely dry).
Subsequently, we calculated the ratio of total days with given grades to the total days of the drought process from the extremely
dry grade to the slightly dry grade. Once the ratio with a given grade first increases to more than 35% of the duration, the
severity of the entire drought process corresponds to this grade.

**Table 1**. Rules of SPI3 grade classification.

| SPI3 value | Grade |
|---|---|
| 0.50 and more | wet |
| -0.49 to 0.49 | near normal |
| -0.99 to -0.50 | slightly dry |
| -1.49 to -1.00 | moderately dry |
| -1.99 to -1.50 | severely dry |
| -2.00 and less | extremely dry |





## 3.2 Divide drought processes according to dry/wet spells

Identified drought processes usually go through one or several dry/wet spells. However, different dry/wet spells usually correspond to various precipitation characteristics and circulation patterns. Basically, it is appropriate to divide drought
processes into different segments and assign these segments into different dry/wet spells. It is beneficial for analysis on drought-related atmospheric and oceanic anomalies during the same dry/wet spells. However, SPI3 on the start date of an identified drought process actually reflects precipitation information in the past 90 days. Therefore, the start date of the drought process used for analysis on atmospheric/oceanic anomalies need to be shifted 90 days in advance, prior to the drought process division.

Using North China as an example, the specified procedures of process division are as follows. Similar to general seasonal classification, we divided the annual period into four dry/wet spells (Table 2) according to the temporal evolution of daily precipitation rate in North China (Fig. 2). It is evident that the wet spell (one-fourth of the annual duration) accounts for over 50% of the total precipitation, while the dry spell (one-third of the annual duration) accounts for about 6%.

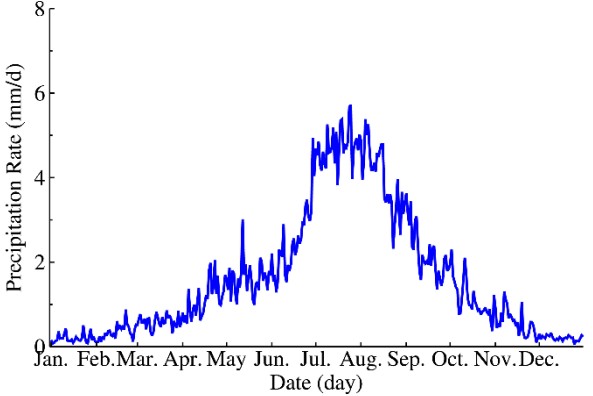

**Figure 2.** Temporal evolution of daily precipitation rate in North China averaged from 1961 to 2010.

**Table 2.** Dates of dry/wet spells and their associated proportions in annual total precipitation in North China. Both Wet–Dry and Dry–Wet represent corresponding transition spells.

| Spell | Period | Precipitation Proportion (%) |
|---|---|---|
| Wet | 21 June–10 September | 56.4 |
| Wet–Dry | 11 September–20 November | 14.9 |
| Dry | 21 November–20 March | 6.3 |
| Dry–Wet | 21 March–20 June | 22.4 |

As illustrated in Fig. 3, we constructed a set of simple rules to divide a drought process into several segments according to dry/wet spells, with the help of Intersection Proportion (IP) and critical Proportion (P, set as 40%). Herein, IP is the proportion





of initial-segment days in days of involved spells. First, we divided one complete process into several initial segments according to dry/wet spells. Secondly, we calculated the IP of initial segments. Third, by comparing IP with P, we assigned these aforementioned initial segments to different dry/wet spells.

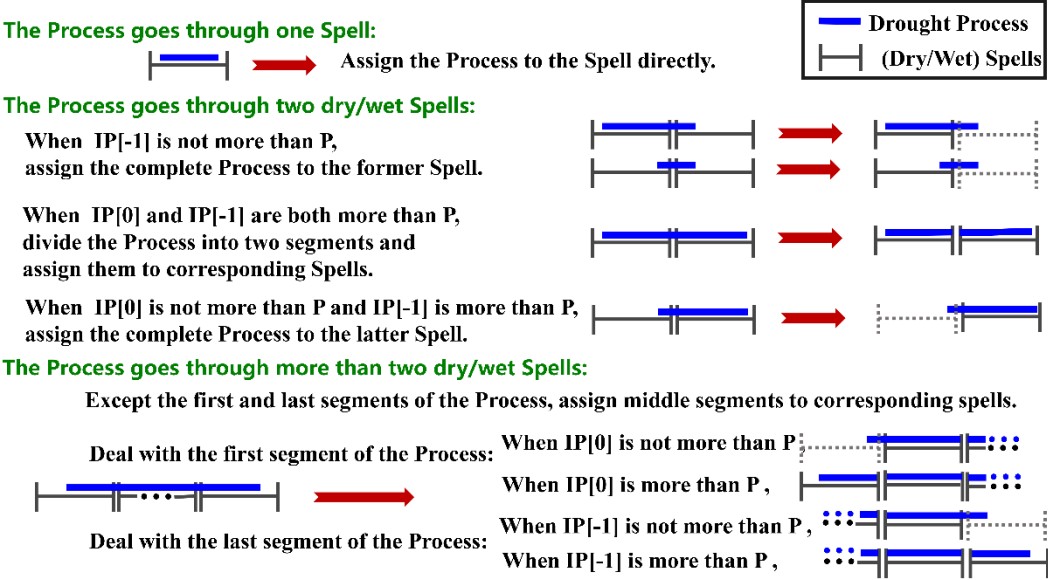

**Figure 3.** Rules of dividing a drought process into segments and assigning them to different dry/wet spells. IP represents Intersection Proportion, while P refers to critical Proportion. The terms "IP[0]" and "IP[-1]" express IP of the former and latter segments respectively, when a drought process is divided into two segments. When it comes to a drought process made up of above two segments, the terms "IP[0]" and "IP[-1]" refer to the first and last IP.

### 3.3 Apply standardized anomalies to identify anomalous atmospheric and oceanic circulation patterns

To identify atmospheric and oceanic anomalies objectively, we chose the method of Standardized Anomalies (SA). It was first used to effectively identify high-impact weather events (Hart and Grumm, 2001;Grumm and Hart, 2001). Subsequently, the method of SA also provided significant values for analysis on extreme precipitation events (Duan et al., 2014;Jiang et al., 2016). Herein, the SA of a meteorological variable was defined by Hart and Grumm (Hart and Grumm, 2001), which is described as

$$SA = \frac{X - \mu}{\sigma},$$

(1)

Where X represents the grid-point value of 200 hPa/500 hPa HGT and SST, while $\mu$ and $\sigma$ are the grid-point mean value and the grid-point standard deviation, respectively. The X used in the SA calculation represents daily variables. Therefore, both of the grid-point $\mu$ and $\sigma$ for the climatological period (1979–2008 for 200 hPa/500 hPa HGT, but 1982–2008 for SST) are also daily variables.



### 3.4 Structure predictors based on the first leading EOF modes of atmospheric and oceanic SA

Because development of a drought process is closely related to spatial-temporal evolution of circulation patterns, it is relatively feasible to build predictors based on these large-scale circulation patterns. Within the same dry/wet spells, we conducted Empirical Orthogonal Function (EOF) analysis on SA during severe and extreme drought process segments respectively.

Furthermore, positive and negative phases in the first leading modes of EOF were used to build predictors. As shown in Fig. 4, a large area of positive phases (B) occurs over the southeast part of China, while a negative center (A) appears to the north of Eurasia. Basically, the predictor is area-averaged over all gridded SA-based variables in selected areas like A and B, considering the positive and negative signals different colors indicate.

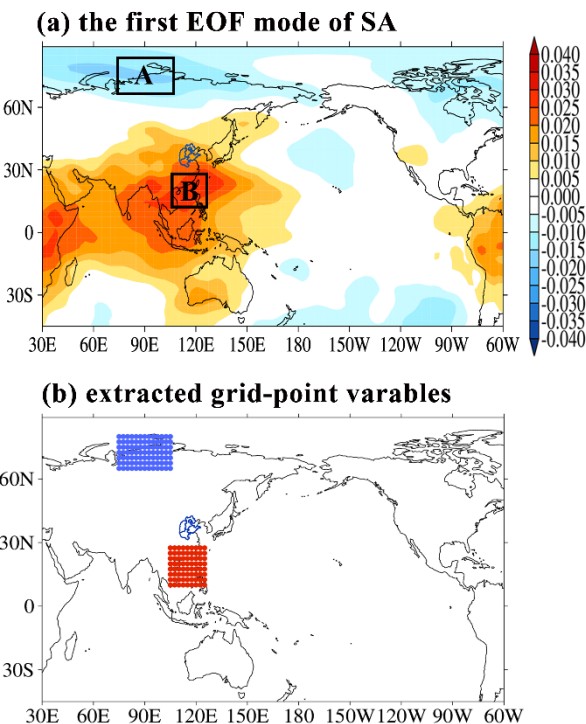

**Figure 4.** An example of how to structure predictors based on the first Empirical Orthogonal Function (EOF) mode of Standardized Anomalies (SA) in North China. Spatial configuration of positive and negative phases is shown in (a), and extracted grid points are shown in (b). Red represents -1, and blue represents 1. Blue solid curves refer to North China. The predictor built is calculated as "B minus A", which is essentially an area-averaged value over all the grid-point SA values after being multiplied by the corresponding signals indicated by the different colors.

### 3.5 Calibrate, validate and operationally use the seasonal drought prediction model

The simple method of stepwise regression was used to build the synchronous statistical relationship between all 90-day-accumulated SA-based predictors and the prediction target SPI3. All the atmospheric and oceanic predictors from all the dry/wet spells were adequately used for model calibration, which reflected drought-related information as integrally as possible.



During the periods of model calibration and validation, this conceptual model is forced by the NCEP/NCAR Reanalysis dataset
(Kalnay et al., 1996).

SPI3 prediction is operationally forced by climate prediction models, which means that the lead-time of this seasonal drought prediction model depends on that of climate prediction models. In our study, CFSv2 (Saha et al., 2014) was operationally used to force seasonal drought process prediction. Prospective 90 day forecasted data subsets of 200 hPa/500 hPa HGT and SST were retrieved from CFSv2, with an interval of about 10 days.


### 3.6 Conduct drought outlook based on angle comparison of prospective 90 day SPI3 curves

Since prospective 90 day SPI3 has been predicted, it is necessary and practical to provide corresponding drought outlook. Rules of drought outlook based on angle comparison of the prospective SPI3 curves were developed in our study (Fig. 5). Generally, positive angles indicate that the current situation tends to be wet, while negative angles represent dry tendencies.
Therefore, two general classes of drought outlook are as follows. (1) When the current condition is no drought (see sketch map I in Fig. 5), the prospective drought situation tends to be no drought or drought occurrence. When the calculated angle $\alpha$ is less than the critical angle $\alpha_1$, the prospective development is drought occurrence; when $\alpha$ is greater than $\alpha_1$, the no-drought situation will persist. (2) Similarly, if the current condition is drought (see sketch map II in Fig. 5), by comparing critical angles $\alpha_2$ and $\alpha_3$, associated drought outlook can be defined: drought persistence ($\alpha$ less than $\alpha_2$), drought recession ($\alpha$ more than $\alpha_2$, but less
than $\alpha_3$), and drought relief ($\alpha$ more than $\alpha_3$).

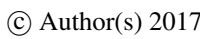



## Current drought condition: No Drought

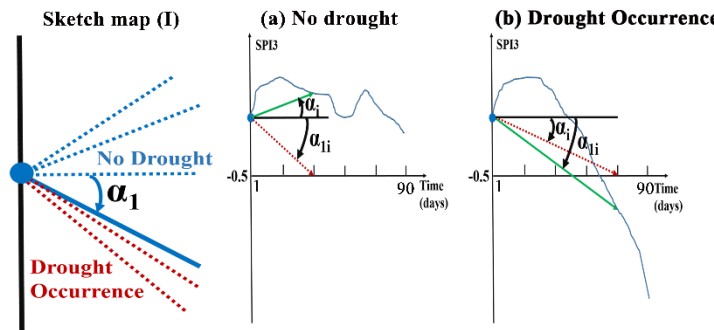

## Current drought condition: In Drought

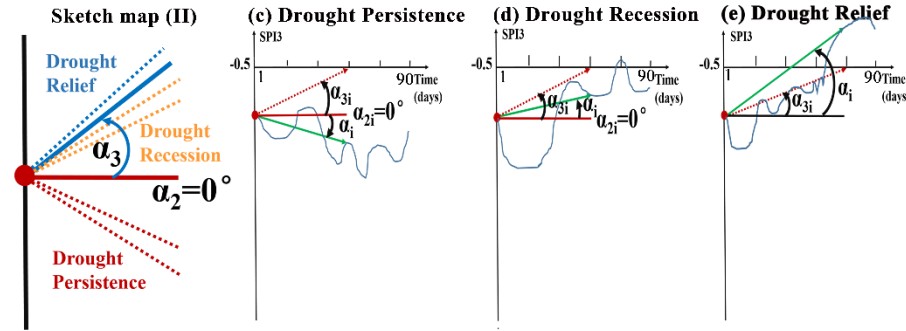

**Figure 5.** Drought outlook based on angle comparison of prospective 90 day SPI3. Sketch maps show examples of current drought conditions for (I) no drought and (II) drought. (a)–(b) and (c)–(e) express different results of drought outlook according to the discrimination rules regarding critical angles in Table 3.

Except the constant critical angle $\alpha_2$ (equal to zero), both $\alpha_1$ and $\alpha_3$ are time-varying, representing angles between the horizontal line and the arrow from the original point (initial prediction time) to the corresponding point on the time axis (see red dashed arrowed lines in Fig. 5(a)–(e)). Similarly, $\alpha_i$ represents angles between the horizontal line and the arrow from the original point to the corresponding point of the predicted SPI3 curve (see green solid arrowed lines in Fig. 5(a)–(e)). Basically, we obtained the comparison results of daily angle series $\alpha_i$ and associated critical angle series $\alpha_{1i}$ ($\alpha_{2i}$ or $\alpha_{3i}$; i=1, 2,..., 90). Based on the

statistical results of angle comparison, we can conduct drought outlook according to rules shown in Table 3.

**Table 3.** Specified rules of drought outlook based on angle comparison. R1 represents the ratio of days when $\alpha_i$ is less than the associated critical angle $\alpha_{1i}$ ($\alpha_{3i}$) to the prospective total 90 days. R2 represents the ratio of days when $\alpha_i$ is greater than the associated critical angle $\alpha_{2i}$ from the 46–90 day to the total 45 days.

| Current SPI3 | Current condition | R1 | R2 | Drought Outlook |
|---|---|---|---|---|
| greater than -0.5 | No Drought | less than 10% | - | No Drought |
| | | greater than 10% | - | Drought Occurrence |
| less than -0.5 | In Drought | greater than 90% | less than 90% | Drought Persistence |
| | | greater than 90% | greater than 90% | Drought Recession |
| | | less than 90% | - | Drought Relief |



## 4 Results

In this section, model construction and calibration were briefly illustrated in Sect. 4.1–4.3, using historical drought events in North China as examples. Then, process simulation and associated drought outlook were illustrated in Sect. 4.4, extending from 2009 to 2014 in North China, East China, and Southwest China. Finally, in Sect. 4.5, recent severe drought processes in these three drought study regions were used to verify the operational application of the conceptual model proposed.

### 4.1 Process Division of seasonal drought events

Following the methodology presented in Sect. 3.1, we have extracted complete seasonal drought processes with severe and extreme grades from the entire SPI3 series during 1979–2008. Identified severe and extreme drought processes in North China are shown in Table 4. Relatively persistent drought periods from 1997 to 2002 in North China are involved, which have also been acknowledged in other associated studies (Rong et al., 2008;Wei et al., 2004). As illustrated in Sect. 3.2, the start date of the drought process was shifted three months in advance, prior to the drought process division. Essentially, the joint complete

process of these drought process segments during different dry/wet spells (Table 5) is slightly distinguished from the identified drought processes (Table 4).

**Table 4.** Identified severe and extreme drought processes from 1979 to 2008 in North China.

| Extreme Drought | 12/6/1997–28/11/1997 |
|---|---|
| | 2/11/1998–11/4/1999 |
| Severe Drought | 15/1/1984–14/5/1984 |
| | 9/11/1988–9/1/1989 |
| | 17/7/1999–1/11/1999 |
| | 23/3/2000–27/6/2000 |
| | 14/4/2001–1/8/2001 |
| | 3/8/2002–4/12/2002 |
| | 26/12/2005–2/2/2006 |

**Table 5.** Drought process segments assigned to dry/wet spells during 1979–2008 in North China.

| Drought Grades | Dry Spell | Dry–Wet Spell | Wet Spell | Wet–Dry Spell |
|---|---|---|---|---|
| Extreme | 21/11/1998–11/4/1999 | 14/3/1997–20/6/1997 | 21/6/1997–10/9/1997 | 11/9/1997–28/11/1997 |
| | - | - | 4/8/1998–10/9/1998 | 11/9/1998–20/11/1998 |
| Severe | 21/11/1983–20/3/1984 | 21/3/1984–14/5/1984 | 21/6/1999–10/9/1999 | 17/10/1983–20/11/1983 |
| | 21/11/1988–9/1/1989 | 18/4/1999–20/6/1999 | 21/6/2001–1/8/2001 | 11/8/1988–20/11/1988 |
| | 24/12/1999–20/3/2000 | 21/3/2000–27/6/2000 | 21/6/2002–10/9/2002 | 11/9/1999–1/11/1999 |
| | 14/1/2001–20/3/2001 | 21/3/2001–20/6/2001 | - | 11/9/2002–4/12/2002 |
| | 21/11/2005–2/2/2006 | 5/5/2002–20/6/2002 | - | 27/9/2005–20/11/2005 |




## 4.2 Predictor Construction

Considering that the development of drought processes is closely related to the spatio-temporal evolution of large-scale circulation patterns, it is feasible that predictors can be constructed based on the first EOF modes of atmospheric and oceanic SA. All the drought process segments during different dry/wet spells were involved in the EOF analysis. As shown in Fig. 6, the spatial configuration of different phases in the 500 hPa HGT fields were adequately considered, including low/high latitude differences (e.g., $P_{HGT500,0}$ in Table 6) and ocean/continent differences (e.g., $P_{HGT500,3}$ in Table 6). Besides, the spatial configuration of different phases surrounding the prediction-targeted region (e.g., Region R/S/T in Fig. 6g) was intentionally used to construct predictors, such as $P_{HGT500,9}$ and $P_{HGT500,10}$ in Table 6. This may be a feasible approach that relates large-scale circulation patterns to the development of drought process effectively and directly. Since the first EOF modes of 200 hPa HGT were similar to those of 500 hPa HGT, the corresponding figures and predictor construction were not presented herein. Additionally, the spatial configuration of different phases in the Pacific SST fields were used, especially in the subtropical gyre zone (Fig. 7 (a)–(d)) and El Niño region (Fig. 7 (e) and (f)). Furthermore, some regions like the El Niño regions R/Q/S were used for construction separately.

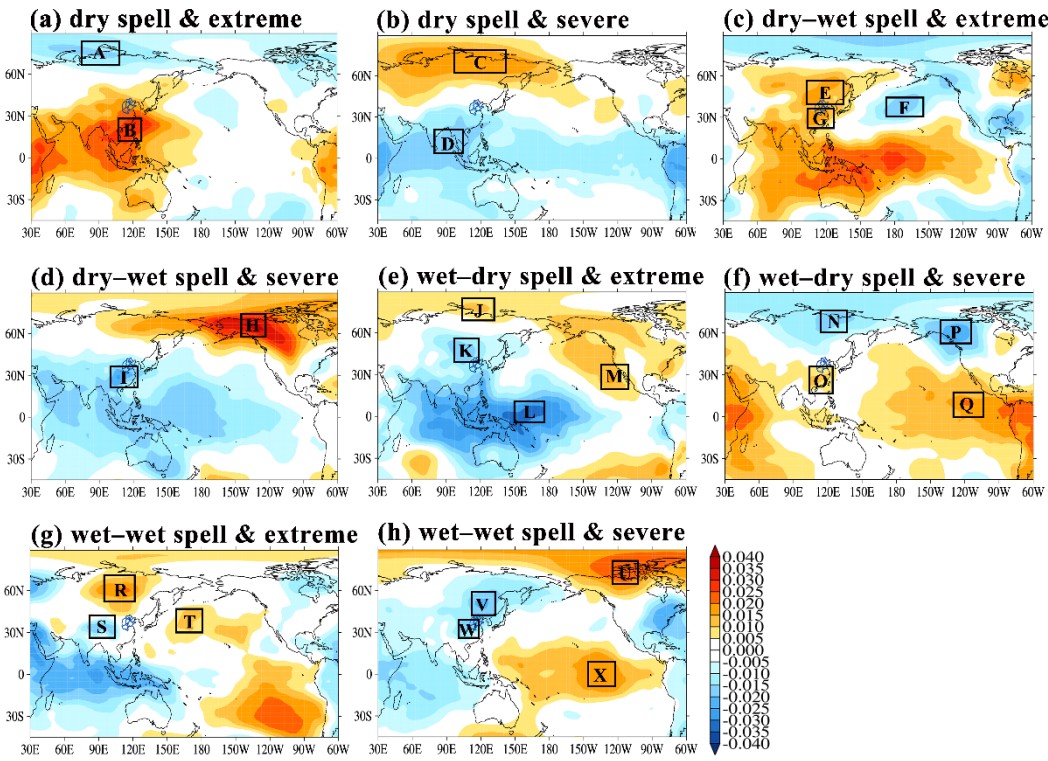

**Figure 6.** The first leading Empirical Orthogonal Function (EOF) modes of Standardized Anomalies (SA) for 500 hPa geo-potential height fields (HGT) during severe and extreme drought process segments in different dry/wet spells. The black boxes represent selected areas that are used to structure predictors, while capital letters refer to the code of selected areas.





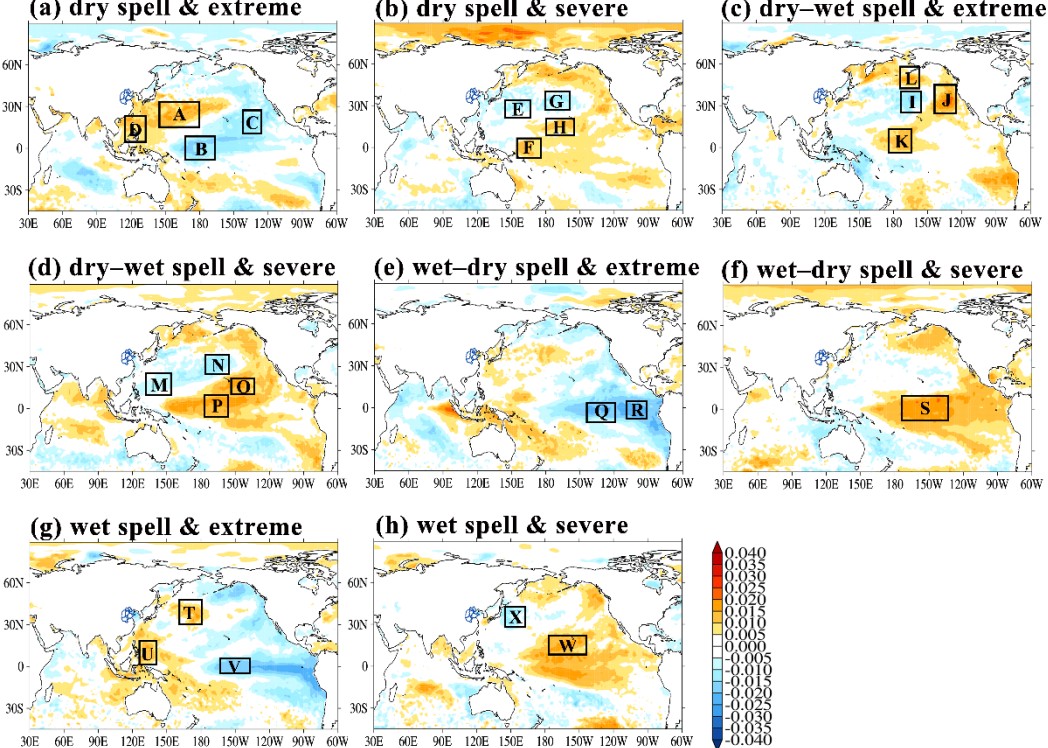

**Figure 7.** Same as Fig. 6, but for Standardized Anomalies (SA) of SST fields.

**Table 6.** Predictor-structured results based on the first leading Empirical Orthogonal Function (EOF) modes for SA of 500 hPa HGT and SST fields during different dry/wet spells in North China. Capital letters refer to the code of selected areas in Fig. 6 and Fig. 7. In the term "$P_{XXX,Y}$", P, XXX, and Y refer to predictor, atmospheric and oceanic elements, and the order of new predictors respectively.

| Dry | Dry–Wet | Wet–Dry | Wet |
|---|---|---|---|
| $P_{HGT500,0}$=B-A | $P_{HGT500,2}$=E-F | $P_{HGT500,5}$=J-K | $P_{HGT500,9}$=R-S |
| $P_{HGT500,1}$=C-D | $P_{HGT500,3}$=G-F | $P_{HGT500,6}$=M-L | $P_{HGT500,10}$=T-S |
| $P_{SST,0}$=A-B | $P_{HGT500,4}$=H-I | $P_{HGT500,7}$=O-N | $P_{HGT500,11}$=U-V |
| $P_{SST,1}$=D-B | $P_{SST,5}$=L+K-I | $P_{HGT500,8}$=Q-P | $P_{HGT500,12}$=X-W |
| $P_{SST,2}$=A-C | $P_{SST,6}$=J-I | $P_{SST,9}$=Q | $P_{HGT500,13}$=U-W |
| $P_{SST,3}$=F-E | $P_{SST,7}$=M-P | $P_{SST,10}$=R | $P_{SST,12}$=T |
| $P_{SST,4}$=H-G | $P_{SST,8}$=N-O | $P_{SST,11}$=S | $P_{SST,13}$=U-V |
| - | - | - | $P_{SST,14}$=W-X |



### 4.3 Model Calibration

The synchronous statistical relationship between SPI3 and all the 90-day-accumulated SA-based predictors from all the dry/wet

spells was calibrated using the simple method of stepwise regression. Six experiments of seasonal drought prediction were

conducted beginning with January 1 of each year (Table 7). Since the calibration period increased year by year, the figure for

samples used for calibration was considerable. Besides, six drought prediction models were statistically significant, and the

corresponding multiple correlation coefficients were no less than 0.75. Statistical parameters showed little change across the

six calibration experiments. Furthermore, calibrated SPI3 curves were almost consistent with the observation data (Fig. 8),

especially with respect to the key turn-points and tendencies.

**Table 7.** Statistical parameters of stepwise-regression equations used for prediction during different calibration periods in North China. All values of F are greater than the corresponding critical values $F_{\alpha=0.05}$, which means that the corresponding equation is statistically significant.

| Calibration period (1 Jan 1983–) | Validation period | Numbers of selected/initial predictors | Multiple correlation coefficient | Value of F | Critical Value of F $F_{\alpha=0.05}$ |
|---|---|---|---|---|---|
| 31 Dec 2008 | 1 Jan 2009–31 Dec 2009 | 38/43 | 0.76 | 337.6 | 1.41 |
| 31 Dec 2009 | 1 Jan 2010–31 Dec 2010 | 37/43 | 0.76 | 352.4 | 1.41 |
| 31 Dec 2010 | 1 Jan 2011–31 Dec 2011 | 39/43 | 0.75 | 345.9 | 1.4 |
| 31 Dec 2011 | 1 Jan 2012–31 Dec 2012 | 39/43 | 0.76 | 370.5 | 1.4 |
| 31 Dec 2012 | 1 Jan 2013–31 Dec 2013 | 38/43 | 0.76 | 389.1 | 1.41 |
| 31 Dec 2013 | 1 Jan 2014–31 Dec 2014 | 39/43 | 0.75 | 375.1 | 1.4 |

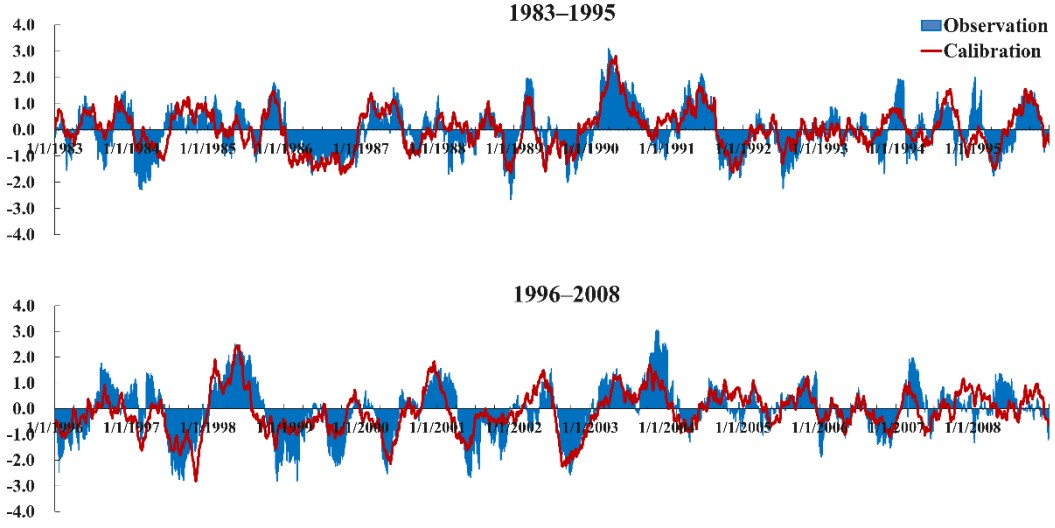

**Figure 8.** Temporal evolution of observed and calibrated SPI3 during the calibration period between 1 Jan 1983 and 31 Dec 2008 in North
China.





## 4.4 Process simulation and outlook forced by the NCEP/NCAR Reanalysis datasets

The synchronous stepwise-regression relationship between all the 90-day-accumulated SA-based predictors and the predictive target SPI3 was calibrated. Essentially, SPI3 simulation can be forced by the NCEP/NCAR Reanalysis datasets. To assess

model performance of severe seasonal droughts, we took recent drought events in Southwest China, East China, and North China as examples. First, Southwest China experienced the 2009/2010 drought and the 2011 summer drought (the black boxes in Fig. 9 (c)). Although the simulated SPI3 did not reach its peak, it indicated the state transformation from drought occurrence to persistence and eventually to relief. In terms of the 2011 summer drought in the Southwest China, the simulated SPI3 indicated that the state remained wet and gradually became wetter, in contrast to the observed drought state. Nevertheless,

during the phase of drought recession, the simulated development was quite similar to the observed development. The simulation of SPI3 performs well in development but is weak in severity. This distinct feature also appears in the simulation of the 2011 drought in East China (the black box in Fig. 9 (b)) and the 2014 drought in North China (the black box in Fig. 9 (a)).

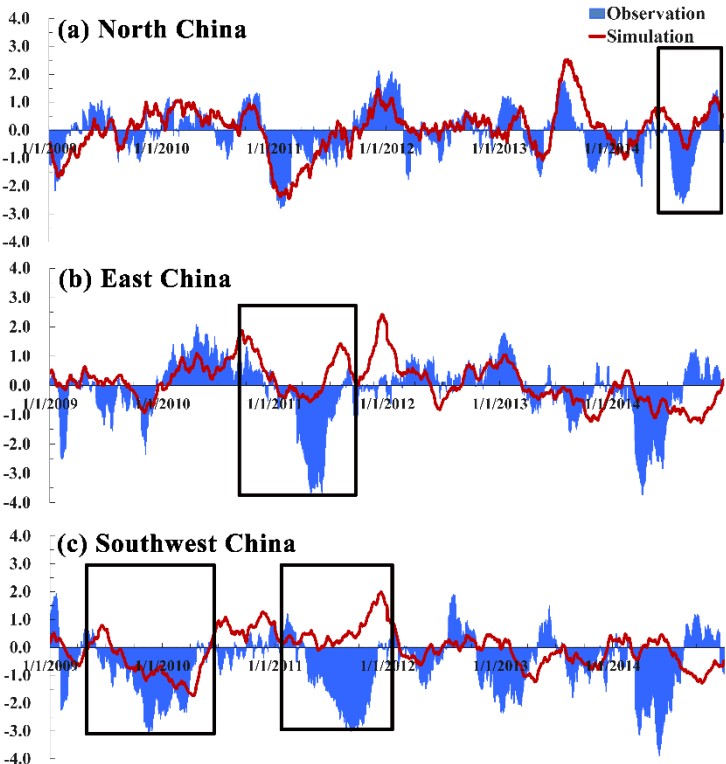

**Figure 9.** Temporal evolution of observed and simulated SPI3 processes during the validation period between 1 Jan 2009 and 31 Dec 2014. The black boxes in (a)–(c) indicate the 2014 summer and autumn drought in North China, the 2011 spring drought in East China, the 2009/2010 drought in Southwest China, and the 2011 summer drought in Southwest China. Red curves refer to simulated SPI3, while curves filled with light blue represent observed SPI3.



Following the method in Sect. 3.6, the prospective 90 day drought outlook was conducted based on angle comparison of the simulated SPI3 curve (Table 8). Similar to operational use, the simulated prospective 90 day SPI3 at every initial prediction time was real-time corrected. In terms of the 2009/2010 drought in Southwest China and the 2011 summer drought in East China, the simulated drought outlook performed well with respect to drought occurrence, persistence, and recession before 2/12/2009 and 1/5/2011 respectively. Simulation of the 2011 drought in Southwest China performed well in August, 2011.

The 2014 summer drought in North China lasted for a relatively short time, resulting in an observed drought outlook that maintained a state of drought relief during the first month of the drought process. Additionally, these four drought outlook remained weak in simulating the development of drought relief after 31/1/2010, 11/5/2011, 11/9/2011, and 21/7/2014, respectively. Weak performance in simulating severity leads to the development of drought recession rather than drought relief.


**Table 8.** Simulation assessment of recent severe drought events in China forced by the NCEP/NCAR Reanalysis datasets. The numbers 0–4 in the above table represent different drought states: No Drought (0), Drought Occurrence (1), Drought Persistence (2), Drought Recession (3), and Drought Relief (4).

| Drought Events | Initial Time | Simul. | Obs. | Asses. | Initial Time | Simul. | Obs. | Asses. | Initial Time | Simul. | Obs. | Asses. |
|---|---|---|---|---|---|---|---|---|---|---|---|---|
| the 2009/2010 drought in Southwest China | 30/6/2009 | 1 | 2 | yes | 28/9/2009 | 3 | 2 | - | 11/1/2010 | 2 | 3 | - |
| | 10/7/2009 | 2 | 2 | yes | 18/10/2009 | 3 | 2 | - | 21/1/2010 | 2 | 3 | - |
| | 20/7/2009 | 2 | 3 | - | 2/11/2009 | 3 | 3 | yes | 31/1/2010 | 3 | 4 | - |
| | 30/7/2009 | 2 | 3 | - | 12/11/2009 | 3 | 3 | yes | 10/2/2010 | 3 | 4 | - |
| | 9/8/2009 | 2 | 2 | yes | 22/11/2009 | 3 | 3 | yes | 20/2/2010 | 3 | 4 | - |
| | 19/8/2009 | 2 | 2 | yes | 2/12/2009 | 3 | 3 | yes | 2/3/2010 | 3 | 4 | - |
| | 29/8/2009 | 2 | 2 | yes | 12/12/2009 | 2 | 3 | - | 12/3/2010 | 3 | 4 | - |
| | 8/9/2009 | 2 | 2 | yes | 22/12/2009 | 2 | 3 | - | 22/3/2010 | 3 | 4 | - |
| | 18/9/2009 | 2 | 2 | yes | 1/1/2010 | 2 | 3 | - | | | | - |
| the 2011 summer drought in East China | 1/1/2011 | 1 | 1 | yes | 2/3/2011 | 1 | 1 | yes | 1/5/2011 | 3 | 3 | yes |
| | 11/1/2011 | 1 | 1 | yes | 12/3/2011 | 3 | 2 | - | 11/5/2011 | 3 | 4 | - |
| | 21/1/2011 | 1 | 1 | yes | 22/3/2011 | 3 | 2 | - | 21/5/2011 | 3 | 4 | - |
| | 31/1/2011 | 1 | 1 | yes | 1/4/2011 | 3 | 3 | yes | 1/6/2011 | 3 | 4 | - |
| | 10/2/2011 | 0 | 1 | - | 11/4/2011 | 3 | 3 | yes | 11/6/2011 | 3 | 4 | - |
| | 20/2/2011 | 1 | 1 | yes | 21/4/2011 | 3 | 3 | yes | 21/6/2011 | 3 | 4 | - |
| the 2011 summer drought in Southwest China | 11/4/2011 | 1 | 1 | - | 1/7/2011 | 3 | 2 | - | 21/9/2011 | 3 | 4 | - |
| | 21/4/2011 | 2 | 2 | yes | 11/7/2011 | 3 | 2 | - | 1/10/2011 | 3 | 4 | - |
| | 1/5/2011 | 2 | 2 | yes | 21/7/2011 | 3 | 2 | - | 11/10/2011 | 3 | 4 | - |
| | 11/5/2011 | 2 | 2 | yes | 1/8/2011 | 3 | 3 | yes | 21/10/2011 | 3 | 4 | - |
| | 21/5/2011 | 4 | 2 | - | 11/8/2011 | 3 | 3 | yes | 1/11/2011 | 3 | 4 | - |
| | 1/6/2011 | 3 | 2 | - | 21/8/2011 | 3 | 3 | yes | 11/11/2011 | 3 | 4 | - |
| | 11/6/2011 | 3 | 2 | - | 1/9/2011 | 3 | 3 | yes | 21/11/2011 | 2 | 4 | - |
| | 21/6/2011 | 3 | 2 | - | 11/9/2011 | 3 | 4 | - | | | | - |





| the 2014 | 1/6/2014 | 4 | 4 | yes | 11/7/2014 | 3 | 3 | yes | 21/8/2014 | 3 | 4 | - |
| summer | 11/6/2014 | 4 | 4 | yes | 21/7/2014 | 3 | 4 | - | 1/9/2014 | 3 | 4 | - |
| drought in North | 21/6/2014 | 4 | 4 | yes | 1/8/2014 | 3 | 4 | - | 11/9/2014 | 3 | 4 | - |
| China | 1/7/2014 | 1 | 1 | yes | 11/8/2014 | 3 | 4 | - | 21/9/2014 | 4 | 4 | yes |

## 4.5 Process prediction and outlook forced by the CFSv2 operationally forecasted datasets

Compared with drought simulation, operationally predicted results may bring some uncertainties into prospective drought processes and drought outlook. As shown in Fig. 10, predicted curves forced by the CFSv2 and CFS products performed slightly worse than the simulated curves forced by the NCEP/NCAR Reanalysis datasets. However, as a whole, the predicted curves at every initial prediction time can also indicate the development of observed drought. For instance, operationally reforecast curves can indicate phases of occurrence, persistence, and relief during the 2009/2010 drought in Southwest China (Fig. 10 (a)). In terms of operational drought outlook (Table 9), operationally predicted results during drought processes in Southwest China and East China were relatively similar to the simulated ones. Simulation and prediction results of drought outlook were different from each other during the first month of the 2014 drought in North China.

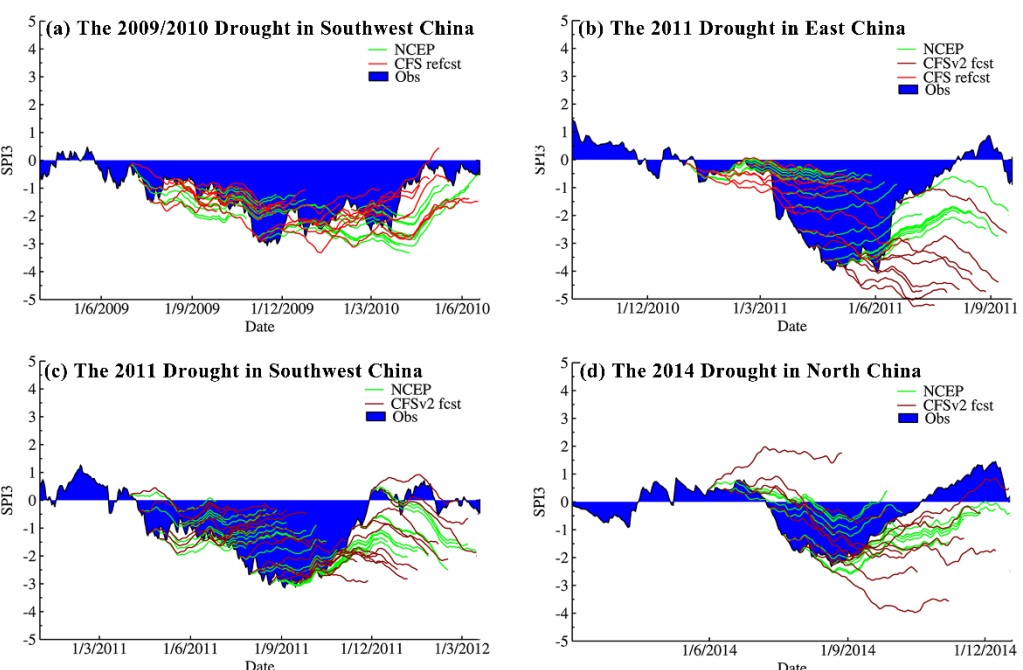

**Figure 10.** Simulation and prediction results of four recent severe drought events in China, corresponding to Table 8 and Table 9. Every unfilled curve represents simulated or predicted prospective 90 day SPI3, with an interval of initial time about 10 days. The curves filled with blue refer to observed SPI3. Dark and bright red curves refer to SPI3 predicted by CFSv2 and CFS products respectively, while light green curves represent SPI3 simulated by the NCEP/NCAR reanalysis datasets.






**Table 9.** Same as Table 8 but for predicted results forced by the operational output from CFSv2.

| Drought Events | Initial Time | Predi. | Obs. | Asses. | Initial Time | Predi. | Obs. | Asses. | Initial Time | Predi. | Obs. | Asses. |
|---|---|---|---|---|---|---|---|---|---|---|---|---|
| the 2009/2010 drought in Southwest China | 30/6/2009 | 1 | 2 | - | 28/9/2009 | 3 | 2 | - | 11/1/2010 | 3 | 3 | yes |
| | 10/7/2009 | 2 | 2 | yes | 18/10/2009 | 2 | 2 | yes | 21/1/2010 | 3 | 3 | yes |
| | 20/7/2009 | 3 | 3 | yes | 2/11/2009 | 3 | 3 | yes | 31/1/2010 | 3 | 4 | - |
| | 30/7/2009 | 3 | 3 | yes | 12/11/2009 | 3 | 3 | yes | 10/2/2010 | 4 | 4 | yes |
| | 9/8/2009 | 2 | 2 | yes | 22/11/2009 | 3 | 3 | yes | 20/2/2010 | 3 | 4 | - |
| | 19/8/2009 | 2 | 2 | yes | 2/12/2009 | 3 | 3 | yes | 2/3/2010 | 3 | 4 | - |
| | 29/8/2009 | 2 | 2 | yes | 12/12/2009 | 3 | 3 | yes | 12/3/2010 | 3 | 4 | - |
| | 8/9/2009 | 3 | 2 | - | 22/12/2009 | 3 | 3 | yes | 22/3/2010 | 3 | 4 | - |
| | 18/9/2009 | 2 | 2 | yes | 1/1/2010 | 3 | 3 | yes | | - | | |
| the 2011 summer drought in East China | 1/1/2011 | 1 | 1 | yes | 2/3/2011 | 1 | 1 | yes | 1/5/2011 | 2 | 3 | - |
| | 11/1/2011 | 1 | 1 | yes | 12/3/2011 | 2 | 2 | yes | 11/5/2011 | 2 | 4 | - |
| | 21/1/2011 | 1 | 1 | yes | 22/3/2011 | 2 | 2 | yes | 21/5/2011 | 2 | 4 | - |
| | 31/1/2011 | 1 | 1 | yes | 1/4/2011 | 2 | 3 | - | 1/6/2011 | 2 | 4 | - |
| | 10/2/2011 | 1 | 1 | yes | 11/4/2011 | 2 | 3 | - | 11/6/2011 | 3 | 4 | - |
| | 20/2/2011 | 1 | 1 | yes | 21/4/2011 | 2 | 3 | - | 21/6/2011 | 3 | 4 | - |
| the 2011 summer drought in Southwest China | 11/4/2011 | 0 | 1 | - | 1/7/2011 | 4 | 2 | - | 21/9/2011 | 3 | 4 | - |
| | 21/4/2011 | 3 | 2 | - | 11/7/2011 | 3 | 2 | - | 1/10/2011 | 3 | 4 | - |
| | 1/5/2011 | 3 | 2 | - | 21/7/2011 | 3 | 2 | - | 11/10/2011 | 3 | 4 | - |
| | 11/5/2011 | 3 | 2 | - | 1/8/2011 | 3 | 3 | yes | 21/10/2011 | 3 | 4 | - |
| | 21/5/2011 | 4 | 2 | - | 11/8/2011 | 3 | 3 | yes | 1/11/2011 | 3 | 4 | - |
| | 1/6/2011 | 4 | 2 | - | 21/8/2011 | 3 | 3 | yes | 11/11/2011 | 4 | 4 | yes |
| | 11/6/2011 | 4 | 2 | - | 1/9/2011 | 3 | 3 | yes | 21/11/2011 | 2 | 4 | - |
| | 21/6/2011 | 3 | 2 | - | 11/9/2011 | 3 | 4 | - | | - | | |
| the 2014 summer drought in North China | 1/6/2014 | 0 | 4 | - | 11/7/2014 | 1 | 3 | - | 21/8/2014 | 3 | 4 | - |
| | 11/6/2014 | 1 | 4 | - | 21/7/2014 | 2 | 4 | - | 1/9/2014 | 4 | 4 | yes |
| | 21/6/2014 | 1 | 4 | - | 1/8/2014 | 3 | 4 | - | 11/9/2014 | 3 | 4 | - |
| | 1/7/2014 | 1 | 1 | yes | 11/8/2014 | 2 | 4 | - | 21/9/2014 | 4 | 4 | yes |

# 5 Discussion

Considering the idea that the development of drought processes is closely related to spatio-temporal evolution of atmospheric and oceanic anomalies, a conceptual prediction model of seasonal drought processes is proposed in our study. Despite its

weakness in prediction of drought severity, the model performs well in simulating and predicting drought development. Since this model proposed is a new attempt, associated issues to discuss are as follows.

First, process prediction and outlook of seasonal drought are the focus in our study. To date, a considerable number of studies have focused on prediction of discrete drought classes (Aviles et al., 2016;Bonaccorso et al., 2015;Chen et al., 2013;Moreira





et al., 2016) and the probability of drought occurrence within certain classes (AghaKouchak, 2014, 2015;Hao et al., 2014).
Compared with these studies, to take seasonal drought processes as the prediction target is another valuable attempt. The conceptual drought model proposed in our study performs relatively well in predicting the development of seasonal drought processes (Fig. 10). Besides, it can indicate drought occurrence, persistence, and relief very well (Table 8 and Table 9), which is meaningful for seasonal water resource management.

Secondly, the model proposed is essentially one stepwise-regression equation, which makes seamless drought prediction for
operational use year-by-year. Despite one simple equation, it involves drought-related spatial and temporal information as integrally as possible. On one hand, precipitation-related synoptic systems appear in the troposphere. Basically, SST, 500 hPa HGT, and 200 hPa HGT are chosen as representatives of the low, middle, upper levels of the troposphere, respectively. On the other hand, all drought process segments assigned to different dry/wet spells are used for EOF analysis within the same dry/wet spells (shown in Sect. 4.2). Predictors are built based on anomaly-related spatial phases of the first EOF modes. Therefore,
adequate drought-related spatio-temporal information has been included in these predictors, which are initially used for model calibration.

Thirdly, the reasons for acceptable performance of operationally predicted results need to be illustrated. Compared with those forced by the NCEP/NCAR Reanalysis datasets (green curves in Fig. 10), the predicted development of drought processes forced by CFSv2 or CFS datasets (red curves in Fig. 10) is relatively similar, especially with respect to the former segment of
every predicted prospective 90 day SPI3 curve. This good performance of operational use benefits from 90-day-accumulated SA-based predictors. It indicates that observed information of atmospheric and oceanic anomalies can be involved to different degrees. For instance, the predicted 90-day-accumulated SA at the prospective $60^{th}$ day is calculated based on a combination of observed data for the past 30 days and forecasted data for the prospective 1–60 day. Therefore, its operational use provides relatively accurate and valuable information. However, it is also worthwhile to investigate how long the predicted drought
processes last is relatively accurate and acceptable, such as the prospective 1–30 day or the prospective 1–60 day. Basically, relevant comparison results with different predicted periods were shown in Fig. 11. It seemed that the 2009/2010 drought in Southwest China and the 2014 drought in North China could be predicted and simulated well even for the prospective 1–75 day. In contrast, the prospective 1–45 day may be a feasible lead time for simulation and prediction of the 2011 droughts in Southwest China and East China, after which the simulated and predicted development changes obviously.





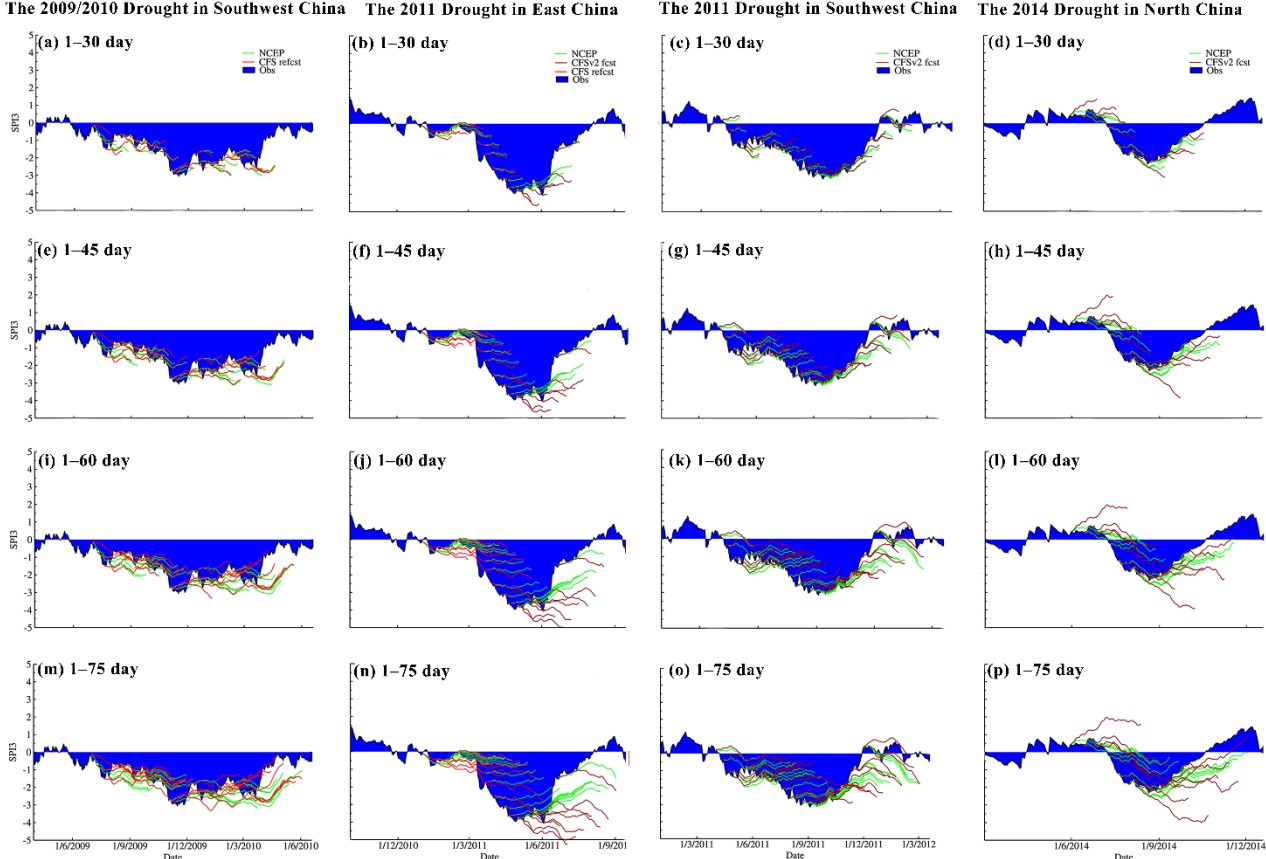

**Figure 11.** Same as Fig. 10 but for different predicted periods, which are namely the prospective (a)–(d) 1–30 day, (e)–(h) 1–45 day, (i)–(l) 1–60 day, and (m)–(p) 1–75 day.

Fourthly, the weak performance in predicting the severity of drought, including the peak point of the drought process and the phase of drought relief, is an important issue. On one hand, it remains weak in predicting the severity of the drought peak. Similar to concluding remarks regarding a probabilistic drought prediction model, this is caused by the typical problem of the inherent averaging effect that depresses the extremes (Behrangi et al., 2015). With the help of real-time correction for operational use, the prediction of drought peaks can be improved to some degree. The other aspect is regarding the prediction of drought relief. As listed in both Table 8 and Table 9, the simulated and predicted results about drought relief were unsatisfying. This weak performance may be associated with precipitation-causing weather patterns during drought relief. They are unsteady and changing dramatically compared with those during drought persistence. Because the period of drought relief is a relatively short phase of the drought process, the relevant information may not be involved in the first EOF modes (Sect. 3.4). Basically, three measures for potential improvement are as follows. (1) More secondary EOF modes, including precipitation-causing circulation patterns during drought relief, can be involved when building initial predictors. (2) The rapid change index (Otkin et al., 2015) could be introduced, to describe temporal changes during drought relief over sub-seasonal





time scales. (3) The empirical scale factor can be introduced to improve drought-relief prediction. The predicted SPI3 during the phase of drought relief could be multiplied by empirical scale factors to strengthen the development of drought relief.

Fifthly, it is necessary to explain the method of predictor construction. The predictor-structured method in our study is somewhat similar to that of tele-connection indices (Wallace and Gutzler, 1981). Nevertheless, it is more goal-directed, because these structured predictors are directly related to synchronous atmospheric and oceanic patterns during different

drought segments within the same dry/wet spells. However, to design geographical ranges of anomalous areas and to combine them was subjective, which led to considerable uncertainties. Accordingly, an objective anomaly-recognized method with explicit critical values needs to be developed. This will contribute to the auto-run feasibility of this conceptual prediction model without artificial interaction.

The sixth issue to illustrate is that synchronous SST anomalies are used in EOF analysis and model calibration. Traditionally,

SST anomalies a few months ahead influence the subsequent regional drought. However, it is also feasible and common that synchronous SST anomalies are used in the investigation of regional drought events in Southwest China (Feng et al., 2014), the Yangtze River basin (Lu et al., 2014), and North China (Wang and He, 2015), which may shape synchronous drought-related circulation patterns. Besides, this is convenient for operational use, while forecasted SST and 200 hPa / 500 hPa HGT can be retrieved together from CFSv2 products at the same initial time.

Finally, the timescales of SA are worthy of being explained. SPI3 reflects comprehensive information regarding the past 90 day accumulated precipitation anomalies. Accordingly, to match the timescale of SPI3, predictors were also calculated based on the past 90-day-accumulated SA-based values, when it comes to model calibration, validation and operational use. However, during the period of predictor construction (Sect. 3.4), EOF analysis was conducted on SA rather than 90-day-accumulated SA-based values. This is also reasonable. If 90-day-accumulated SA-based values were used in EOF analysis, anomalous

information 90 days before the start date of drought processes would also be included in the first EOF modes, which weaken the roles of anomalies during identified drought processes.

## 6 Conclusions

Drought prediction is fundamental for seasonal water management. In this study, we constructed a conceptual prediction model of seasonal drought processes based on synchronous Standardized Anomalies (SA) of 200 hPa/500 hPa geo-potential height

(HGT) and sea surface temperature (SST), considering the idea that drought development is closely related to the spatial-temporal evolution of large-scale circulation patterns. This model can be used for seamless drought prediction and drought outlook, forced by seasonal climate prediction models. We used North China as an example for methodology introduction and used four recent severe drought events in China for application. The main process is as follows. (1) 3-month SPI (SPI3) updated everyday was used to capture severe and extreme drought processes. (2) Empirical Orthogonal Function (EOF) analysis was

applied to SA of 200 hPa/500 hPa HGT and SST during drought process segments within the same dry/wet spells. Subsequently, spatial configurations of the first EOF modes were used to structure SA-based predictors. (3) The synchronous stepwise-





regression relationship between SPI3 and all 90-day-accumulated SA-based predictors were calibrated using the NCEP/NCAR reanalysis datasets. (4) To achieve prospective 90 day drought outlook, we further developed an objective method based on angle comparison of the predicted prospective 90 day SPI3 curves. (5) Eventually, simulation and prediction of seasonal

drought processes, together with drought outlook, were forced by the NCEP/NCAR reanalysis datasets and the NCEP Climate Forecast System Version 2 (CFSv2) operationally forecasted datasets, respectively. Model application during four recent severe drought events in China reveals that the model is good at development prediction but weak in severity prediction. It indicates that the conceptual drought prediction model proposed in our study is potentially another valuable addition to current researches on drought prediction.

**Acknowledgements**

This work is supported by the Special Public Sector Research Program of Ministry of Water Resources (Grants No. 201301040 and 201501041), the Fundamental Research Funds for the Central Universities (Grant No. 2015B20414), the Program for New Century Excellent Talents in University (Grant No. NCET-12-0842), the National Natural Science Foundation of China (Grant No. 51579065), and the Natural Science Foundation of Jiangsu Province of China (Grant No. BK20131368).

**Competing interests**

The authors declare that they have no conflict of interest.

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
