# Peer review of "A conceptual prediction model for seasonal drought processes using atmospheric and oceanic Standardized Anomalies and its application to four recent severe drought events in China"

_Hydrology and Earth System Sciences, 2017_

## Referee Comment (RC1) · Anonymous Referee #1 · 22 Apr 2017

The manuscript illustrates a prediction model of sesonal droughts based on atmospheric/oceanic standard anomalies (SA). In particular, the model is based on synchronous relationship between SPI3 and 90-day accumulated SA anomalies.

Although the paper addresses an interesting topic within the scope of the journal, by proposing a novel methodology, I believe it cannot be published in its current form. My main criticisms are related to the fact that the proposed methods are poorly described or are unclear in several parts of the manuscript.

Major comments follow: - in Section 3.1, details on SPI computation (which seems to

be different from the approach originally proposed by McKee et al., 1993) are lacking ; - in Section 3.2, division of drought processes is rather obscure. Why do you need to split years in dry/wet periods? SPI is computed on a 90-day period, but some of the identified spells (see table 2) cover a shorter period. How do you deal with this issue? What do you mean with initial-segment days (see lines 125-129)? Figure 3 is unintelligible. - in Section 3.6, the description of the angle comparison approach is rather messy. Please clarify and check the correctness of mathematical notations (i.e. subscripts of the angles). What is R2 in Table 3 and how is it calculated? - in Section 4.1, please add further information on the content of Table 5. - in Section 4.3, the model calibration procedure is also ambiguous What is F in Table 7? Please provide a list of the initial 43 predictors and the selected ones. - in Section 4.4, the synchronous stepwise-regression relationship should be described in-depth.

Overall, the lack of clarity in the methodology makes difficult to verify the quality of the derived results.

Finally, I would also suggest the authors to revise the language of the manuscript in order to make it more fluid and comprehensible.

---

## Referee Comment (RC2) · Anonymous Referee #2 · 2 May 2017

This paper proposes a statistical drought prediction model based on atmospheric and oceanic variables. The authors first identify severe and extreme drought events based on the SPI3 and identify predictors for these events. Based on these, they build a drought prediction model and propose a drought outlook. The performance of the full chain is then illustrated in the case of four drought events in China.

*General comment*

I believe that this paper is a valuable contribution to the special issue. However, I believe that, in its current form, it is hard for the reader to follow and process the large

amount of information it contains. For clarification, I would suggest reorganizing the paper. Indeed, some of the subsections in the Methods section bring little to the paper in their current state (especially subsections 3.4 and 3.5). I could suggest two ways (non-restrictive) to reorganize the Methods and Results sections. (1) The first suggestion would be to keep the current structure but making sure that the Methods section (a) is more detailed and explains even briefly all methods, including the computation of the SPI, the step-wise regression and the EOF analysis, and (b) excludes statements on what has been done (move to the Results section). (2) The second way could be to separate the paper by "themes" or "work steps" as listed at the end of the introduction: this way, the continuity between the steps could be easier to follow, and, for instance, the drought periods and predictors would be available to the reader to understand the steps of "structuring predictors" and "building the prediction model".

\*Major comments and general questions\*

- Introduction: Even if it becomes clear early in the paper, I think it should be stated that the droughts studied are restricted to meteorological droughts.

- Section Methods: I was missing descriptions of the computation of the SPI, the EOF analysis, as well as of the step-wise regression used to build the prediction model. These could simply be described in very brief sentences.

- Lines 112-114: Could you please explain why you chose the first date of the period as the beginning for the drought period? Couldn't that lead to overestimating the duration of the droughts, and subsequently influence the selection/use of predictors?

- Line 142: Are these the circulation pattern variables used in the building of the model? If so, it could be worth emphasizing them throughout the Methods section when appropriate.

- Lines 148-150: in my opinion, these lines state analyses that have been carried out and do not really inform on the methodology itself. A brief sentence describing the

[Figure]

EOF analysis could be useful here. Knowing the severe and extreme drought process segments at this stage could help towards a more pragmatic description of the method.

- Lines 162-163 (also see previous comment): The sentence "All the atmospheric and oceanic predictors from all the dry/wet spells were adequately used for model calibration, which reflected drought-related information as integrally as possible." does not seem to be supported by anything at this stage. I would suggest moving it to the Results section if appropriate, or reformulating the sentence.

- Section 3.6: I would have liked the authors to explain the advantage of this method over the methods found in the literature. In addition, I think this subsection needs some clarifications.

- Figure 8: could you please further detail the legend for Table 8? I believe "above table" should be changed to below. Could you describe what should be read in each column? More specifically, the column "Asses." seems to indicate when the simulation and observation agree. If this is correct, the "yes" entry for 30/6/2009 should be "-", and the "-" for 11/4/2011 should be "yes".

- Lines 287-288: Is this observation based on a visual inspection of Figure 10?

- Tables 8 and 9: It seems that the prediction model performs better when forecasting the 2009/2010 drought in Southwest China than in simulating it. Why do you think this happens?

*Minor comments*

- Throughout the paper, citations were sometimes organized based on alphabetical order and sometimes based on year of publication. These should be consistent.

- L.32: The full name of SPI is "Standardized Precipitation Index".

- L.69: Please explain the abbreviation "SA", as it has not been explained before in the text (only in the abstract).

- Section 3 Methods: I would recommend changing the titles of subsections 3.1 to 3.6. The titles should reflect what is presented in the sections, i.e. here methods and techniques, and therefore should avoid action verbs (using, divide, apply,...). In my opinion, action verbs can be misleading and can make the reader expect results.

- Lines 147 and 303: "spatial-temporal" and "spatio-temporal" are used in these two sentences.

---

## Short Comment (SC1) · 10 May 2017

Dear Referee#1,

Thank you for pointing out problems of this manuscript, especially some implicit expression and loss of relatively key information. This manuscript is under the careful revision. After reading your comments carefully, I still have two places to consult. Here they are:

(1)"- in Section 4.4, the synchronous stepwise-regression relationship should be described in-depth."

Do you mean the illustration about the specified stepwise-regression equation needs to be provided? Personally, it is described in the form of model calibration in section 4.3. Therefore, I am a little confused about how to describe it in-depth in section 4.4.

(2)"please add further information on the content of Table 5."

Table 5 is the process-split result in North China, since drought processes in North China are used as example of methodology description. I am unclear about what the further information refers to. Since droughts in East China and Southwest China are also used in model application, does the further information refer to the process-split results in East China and Southwest China?

These are the two issues to consult. Additionally, I really thank you for pointing out these problems, which inspire authors to think about how to make readability improved a lot.

---

## Referee Comment (RC3) · Anonymous Referee #1 · 14 May 2017

Dear Dr Liu

with reference to the stepwise regression, I suggest to better clarify:

- the structure of the multiple regression models (linear or not?); - the explained variables (the first PCs of SA predictors reported in Table 6?) and which criterion is used to select the most significant ones; - how the calibration and validation periods have been chosen (see Table 7) and which of them is finally applied.

With respect to the second issue, I guess that Table 5 is coherent with Table 1 (drought classification) and Table 2 (division of annual period). What is unclear to me is the

division of drought process (initial segment days?) illustrated at lines 125-129 and in Figure 3 and its application.

I hope these comments could help you in the revisionn of the manuscript.
* * *

---

## Short Comment (SC2) · 14 May 2017

Dear Referee#1,

Thank you for the supplementary illustrations about these two comments very much. I have fully realized the problems of insufficient and unclear illustrations. I, together with co-authors, will make adequate works of revision to improve its clarity and readability.

Best Regards,

Zhenchen Liu

---

## Author Comment (AC1) · 24 May 2017

**Response to comments of Anonymous Referees**

Our responses to the referee's comments are shown below in blue, with the reviewer's comments shown as normal text.

**Response to comments of Anonymous Referee #1 at the round1**

**Overall assessments:**

The manuscript illustrates a prediction model of seasonal droughts based on atmospheric/oceanic standard anomalies (SA). In particular, the model is based on synchronous relationship between SPI3 and 90-day accumulated SA anomalies.

Although the paper addresses an interesting topic within the scope of the journal, by proposing a novel methodology, I believe it cannot be published in its current form. My main criticisms are related to the fact that the proposed methods are poorly described or are unclear in several parts of the manuscript.

RESPONSE:

Thank you for your feedback about this manuscript. Actually, the synchronous predictor-SPI3 statistical relationship forced by dynamical products, together with process prediction, are new and valuable attempt in the field of drought prediction. Besides, the process prediction model performs well at predicting seasonal drought development, despite its weakness in predicting drought severity. It is also an important result. As a whole, the paper actually addresses an important topic with a novel methodology.

Since it is a complete drought process prediction model, the procedure of model construction contains adequate but necessary information. Although we tried our best to illustrate it, the original manuscript still lack clarity. With comments you and Referee#2 made, we have realized the problems to solve. Large amounts of work are being conducted to improve it, especially in the structure of the manuscript. In the potentially revised version, we will give up the expression pattern of methodology and result section. Instead, we will choose the "theme-workstep" pattern for clarity, which is the comment Referee#1 made. By doing so, the continuity between the steps could be easier to follow. For example, a flow diagram map of model construction will be inserted in the end of the Introduction section. Accordingly, a brief but general introduction about the sequential procedures will also be given. Sections and sub-sections will be adjusted, following the sequential procedures of model construction. Additionally, brief but necessary

text description, tables and figures will be added in the feasible position. Basically, we hope the quality of the manuscript will be improved as much as possible, and it can be more readable and easily understood.

30

**Major comments:**

**- in Section 3.1,**

details on SPI computation (which seems to be different from the approach originally proposed by McKee et al., 1993) are lacking;

35 RESPONSE:

Thank you for pointing out this problem. We will add a flow chart to illustrate the steps of calculating SPI3 updated everyday in detail. Besides, we have also made text description clear and simple. The revised text description and flow chart are shown as follows: "*SPI3 was used as the drought index for seasonal drought recognition and prediction in this study, and the period for SPI3 calculation is 1979–*
40 *2014. Traditionally, the SPI3 set is moving in the sense that each month a new value is determined from the previous 3 months (McKee and Kleist, 1993). To obtain seasonal drought processes at the one-day timescale, we chose to update SPI3 everyday, which was also recommended by the World Metrological Organization (2012). Compared with the traditional method, the essential difference is that the interval for SPI3 calculation has been extended from 12 months to 365 days, while the moving window has*
45 *changed from one month to one day. However, no changes happen to relevant mathematic procedures. Specified illustrations and details about how to calculate SPI3 updated everyday are shown as Fig. 2.*"

[Figure]

**Figure 2.** Illustration of calculating SPI3 updated everyday. The letter "E" represents value existence, while the letter "N" represents no relevant data.

50

**- in Section 3.2,**

division of drought processes is rather obscure. Why do you need to split years in dry/wet periods? SPI is computed on a 90-day period, but some of the identified spells (see table 2) cover a shorter period. How do you deal with this issue? What do you mean with initial-segment days (see lines 125-129)? Figure 3

55  is unintelligible.

RESPONSE:

Thank for your valuable and advisable feedback, which help me realize the problems and make the description clearer. Corresponding responses are organized as follows:

 (1) Why to split years into dry/wet periods

60  RESPONSE: Essentially, it serves the following step of predictor construction, in which drought-related atmospheric/oceanic anomalies within the same dry/wet spells are extracted and used for anomaly-based predictor construction. The main reason is that drought-related circulation patterns during different dry/wet periods are different. As illustrated in Lines 108-111 in the original version, one complete drought process usually goes through one or several dry/wet spells. Different dry/wet spells usually correspond to

various precipitation characteristics and circulation patterns. Therefore, it is appropriate to divide drought processes into different segments and assign these segments into different dry/wet spells.

(2) "SPI is computed on a 90-day period, but some of the identified spells (see table 2) cover a shorter period."

RESPONSE: Actually, connections among timescale of SPI3, drought processes and dry/wet spells need to be illustrated indeed. SPI is computed on a 90-day period (SPI3), used to identify seasonal (90-day timescale) drought processes. Dry/wet spells are used to split identified complete drought processes. However, timescale of SPI3 and dry/wet spells have no relationship with each other. We think that the cause of confusion lies in the originally implicit description about SPI3 calculation and its application in seasonal drought process identification. In the potentially revised version, the explicit description and two feasible sketch maps will be provided.

(3) the expression of initial-segment days

RESPONSE: Initial segments are actually the split drought process segments according to dry/wet spells, which are used to compute Intersection Proportion (IP). The previous description about these two terms are confusing. In the revised paper, we will replace "Herein, IP is the proportion of initial-segment days in days of involved spells." with the new expression "*Herein, IP is the proportion of initial segments accounting for relevant dry/wet spells, and the initial segments (e.g., D1, D3 and D4 in Fig. 6) refer to parts of one drought process split by dry/wet spells*". Additionally, relevant sketch maps will be provided for clarity.

(4) Figure 3 is unintelligible

RESPONSE: The original expression is implicit and unintelligible indeed. We think two places need to be revised. On one hand, it lacks calculation expression of IP[0] and IP[-1]. In the revised version, we will take the processes going through two dry/wet spells as an examples, and add the simple information of IP[0] and IP[-1] calculation. On the other hand, the original sketch maps about processes going through more than two dry/wet spells lack the information before assignment. In the revised version, this problem will be solved. The revised figure is shown as follows.

[Figure]

**Figure 1.** Process-split rules of one drought process according to dry/wet spells. IP represents Intersection Proportion, while P refers to critical Proportion. The terms "IP[0]" and "IP[-1]" express IP of the start and end segments respectively.

**- in Section 3.6,**

the description of the angle comparison approach is rather messy. Please clarify and check the correctness of mathematical notations (i.e. subscripts of the angles). What is R2 in Table 3 and how is it calculated?

RESPONSE:

Actually, the implicit description of the angle comparison approach is also pointed out in the comments Referee#2 made. In the potentially revised version, we will transform the original text into three sub issues. They are namely "how to describe drought development", "general classifications of drought outlook" and "how to calculate angles and conduct angle-based drought outlook". By doing so, we hope that it can be explicit and easily understood.

We have checked mathematical notations (i.e. subscripts of the angles). Currently, we have found a mathematical notation in the caption of original Table 3. We have changed "$\alpha_i$ is greater than critical angle $\alpha_{2i}$" into "$\alpha_i$ is greater than critical angle $\alpha_{3i}$". Besides, the original description about mathematical notations is confusing indeed. In the revised version, it has been clarified in the sub issue of "how to calculate angles and conduct angle-based drought outlook".

Finally, R2 represents the ratio of specific days in the period of the predicted prospective 46–90 days. These specific days meet the criteria that $\alpha_i$ is greater than critical angle $\alpha_{3i}$. It is dividing selected specific

days by 45 (the 46$^{th}$ - 90$^{th}$ day) days, which can be found in the definition of R2. Illustration about R2 can be found in the caption of original Table 3.

**- in Section 4.1,**

please add further information on the content of Table 5.

[Supplementary comment: I guess that Table 5 is coherent with Table 1 (drought classification) and Table 2 (division of annual period). What is unclear to me is the division of drought process (initial segment days?) illustrated at lines 125-129 and in Figure 3 and its application.]

RESPONSE:

Thank you for this feedback. Actually, results in original Table 5 follow the division rules in original Figure 3 and the dry/wet spells in original Table 2. However, it is incoherent with original Table 1. (Table 1 is used to assign dry/wet grades to every daily SPI3 value of an identified complete drought process. It will be further illustrated in the potentially major version.)

We have to admit that the original division rules itself (original Figure 3) are actually confusing. In the potential revision, we will add a new figure (Figure 7.) as an intermediate result of drought process division. Additionally, detailed but necessary labels will be inserted into the feasible positions in original Figure 3, which help people understand the process division rules. The following figure is the intermediate sketch map.

 **Figure 2.** Comparison results of P, "IP[0]" and "IP[-1]" of drought processes during 1979–2008 in North China. The start dates of these drought processes have been shifted 90 days in advance. IP represents Intersection Proportion, while P refers to critical Proportion. The terms "IP[0]" and "IP[-1]" express IP of the start and end segments respectively, when a drought process is divided into two segments.

**- in Section 4.3,**

the model calibration procedure is also ambiguous What is F in Table 7? Please provide a list of the initial 43 predictors and the selected ones.

RESPONSE:

Actually, in original Table 7, F represents values of the constructed F-test statistic in the final stepwise regression, while $F_{\alpha=0.05}$ refer to F-test critical values at 5% significance level. Considering F and $F_{\alpha=0.05}$ are included in the details of procedures, we tend to delete these two columns of F and $F_{\alpha=0.05}$ in original Table 7 in the potentially revised version.

Besides, a list of the initial 43 predictors and the selected ones will be shown in the potentially revised version. Since all the initial 43 predictors are shown, we will also add the information of predictor construction in the 200 hPa HGT field into the manuscript. Accordingly, the first leading EOF modes of SA for 200 hPa HGT will also be shown.

**- in Section 4.4,**

the synchronous stepwise-regression relationship should be described in-depth.

[I suggest to better clarify:

- the structure of the multiple regression models (linear or not?);

- the explained variables

(the first PCs of SA predictors reported in Table 6?) and which criterion is used

to select the most significant ones;

- how the calibration and validation periods have been chosen (see Table 7) and which of them is finally applied.]

RESPONSE:

Thank you for pointing out this problem about clarity. Stepwise regression is also multiple and linear. Essentially, stepwise regression selects a best subset of explanatory variables used for model construction

while all the explanatory variables are used to build multiple linear regression model. Additionally, positive and negative pattern areas on the first leading EOF modes are used to build SA predictors in our study. SA predictors reported in original Table 6 is actually explanatory variables, instead of the first PCs (Principal Component) of SA predictors.

The general description about criterion used to select the most significant ones (Afifi and Azen, 1972) is as follows. "The technique of stepwise regression which selects a best subset according to the following procedure: The first step selects the single variable which best predicts Y. The second step finds the variable which best predicts Y given the first variable entered. In the steps that follow, either: (a) a variable is entered which best improved the prediction of Y given all the variables entered from the previous steps; or (b) a variable is removed from the set of predictors if its predictive ability falls below a given level. The process is terminated when no further variable improves the prediction of Y".
However, these aforementioned information is detailed. Considering large amounts of information this manuscript contains, we tend to give a brief but important introduction about the structure, explanatory variables and criterion of stepwise regression in the potentially revised version. Additionally, the citation of the introduction about stepwise regression (Afifi and Azen, 1972) will be also provided.
References:
Afifi, A. A., and Azen, S. P.: Statistical analysis: a computer oriented approach, Academic press, 1972.

In terms of selection approaches of calibration and validation periods, two reasons are responsible for it. First, we want to examine model performance on the 2009/2010 Southwest China drought. Therefore, the end of the first calibration period is 31 December 2008. Second, consistent with the drought prediction year by year, the calibration period is running and extended to the day before the new year. For example, the seasonal drought prediction model calibrated from 1 Jan 1983 to 31 Dec 2011, is used for everyday initial prediction time in the whole year of 2012. When it comes to every initial drought prediction in the year of 2013, the corresponding drought model is calibrated from 1 Jan 1983 to 31 Dec 2012. Accordingly, every SPI3 value in original Fig.9 and original Fig.10 is simulated or predicted, using the drought prediction model with corresponding calibration period. We think the original description in original

section 4.3 is unclear and confusing. In the potentially revised version, we will make a brief but necessary explanation about the selection of calibration.

190

Overall, the lack of clarity in the methodology makes difficult to verify the quality of the derived results.
RESPONSE:
Thank you for pointing out the lack of clarity. Actually, large amounts of revision work is being conducted
195 to improve it. For example, a general flow diagram of model construction will be provided at the end part of the Introduction section, aiming at a brief instruction about sub-sections. Besides, the original Figure 3 will also be readable, by means of adding some necessary labels. As an important revision designed for the potentially revised version, **general introduction to the sequential procedures of model construction** is as follows:

200 **(1) Descriptive texts in the end of the Introduction section**

*"Considering that the conceptual model proposed consists of several important parts, a brief but general introduction about sequential procedures are shown (Fig. 1), prior to specified illustration from sect. 3 to sect. 8. In sect. 3, historical extreme and severe historical drought processes will be identified with 3-month SPI updated everyday (SPI3). Identified drought processes usually go through one or several*
205 *dry/wet spells, in which precipitation deficit characteristics and circulation patterns varies. Therefore, process-split rules according to dry/wet spells in sect. 4 are designed to assign drought process segments to different dry/wet spells. Meanwhile, gridded values in the fields of 200 hPa/500 hPa HGT and SST are transformed into gridded values of Standardized Anomalies (SA) in sect. 5. Basically, maps of atmospheric/oceanic SA during drought process segments within the same dry/wet spells are the*
210 *important inputs of predictor construction. After Empirical Orthogonal Function (EOF) analysis are conducted on these SA-based maps, the first leading EOF modes are used to build up predictors (sect. 5). Further, synchronous statistical relationship between SA-based predictors and SPI3 are calibrated with the method of stepwise regression in sect. 6. The National Centers for Environmental Prediction / National Center for Atmospheric Research (NCEP/NCAR) Reanalysis datasets and the NCEP Climate*
215 *Forecast System Version 2 (CFSv2) operationally forecasted datasets are used to force the synchronous*

*statistical relationship, respectively. Simulated and predicted 90-day prospective SPI3 time series are output of sect. 7. With the help of angle-based rules of drought outlook, simulated and predicted SPI3 time series are transformed to five kinds of drought outlook, which are easily accessible to end water managers."*

220 **(2) General flow chart**

[Figure]

**Figure 3.** Brief introduction about sequential procedures of the drought prediction model construction

**(3) Section assignments**

3 Identification of drought processes

225      3.1 Three-month SPI updated everyday

     3.2 Drought process identification and grade classification

4 Drought process division according to dry/wet spells

5 Predictor construction

     5.1 Atmospheric and oceanic standardized anomalies

230      5.2 The first EOF leading modes of SA

     5.3 Pattern-based predictor construction

6 Model calibration

     6.1 Synchronous statistical relationship

     6.2 Rolling calibration year by year

235 7 Drought process simulation and prediction

     7.1 Model forcing

7.2 Drought processes simulated by the NCEP/NCAR reanalysis datasets

7.3 Drought Processes predicted by the CFSv2 forecast datasets

Finally, I would also suggest the authors to revise the language of the manuscript in order to make it more fluid and comprehensible.

245 RESPONSE:

Thank you for this comment. We have followed almost all the comments, and this manuscript is being revised as much as possible by ourselves. Once finished, we will further invite professional editors at Editage, a division of Cactus Communications to revise it and improve the language quality. We try our best to make this manuscript more fluid, readable and comprehensible.

250

---

## Author Comment (AC2) · 24 May 2017

**Response to comments of Anonymous Referees**

Our responses to the referee's comments are shown below in blue, with the reviewer's comments shown as normally black text.

**Response to comments of Anonymous Referee #2 at the round1**

This paper proposes a statistical drought prediction model based on atmospheric and oceanic variables. The authors first identify severe and extreme drought events based on the SPI3 and identify predictors for these events. Based on these, they build a drought prediction model and propose a drought outlook. The performance of the full chain is then illustrated in the case of four drought events in China.

*General comment*

I believe that this paper is a valuable contribution to the special issue. However, I believe that, in its current form, it is hard for the reader to follow and process the large amount of information it contains. For clarification, I would suggest reorganizing the paper. Indeed, some of the subsections in the Methods section bring little to the paper in their current state (especially subsections 3.4 and 3.5). I could suggest two ways (non-restrictive) to reorganize the Methods and Results sections. (1) The first suggestion would be to keep the current structure but making sure that the Methods section (a) is more detailed and explains even briefly all methods, including the computation of the SPI, the step-wise regression and the EOF analysis, and (b) excludes statements on what has been done (move to the Results section). (2) The second way could be to separate the paper by "themes" or "work steps" as listed at the end of the introduction: this way, the continuity between the steps could be easier to follow, and, for instance, the drought periods and predictors would be available to the reader to understand the steps of "structuring predictors" and "building the prediction model".

RESPONSE:

Thanks for your admiration about the scientific values of this manuscript. Actually, it is a new and valuable attempt of seasonal drought process prediction, which hardly appear in the previous study.

The most important issue to solve is the lack of clarification, especially in the Methods section and Results section. In the potentially revised version, we tend to choose the second way recommended, which separate the paper by "themes" or "work steps".

To achieve it, we will add a flow diagram map of model construction at the end of the Introduction section and give a brief introduction about the sequential procedures. Here they are. "*Considering that the conceptual model proposed consists of several important parts, a brief but general introduction about sequential procedures are shown (Fig. 1), prior to specified illustration from sect. 3 to sect. 8. In sect. 3, historical extreme and severe drought processes will be identified with 3-month SPI updated everyday (SPI3). Identified drought processes usually go through one or several dry/wet spells, in which precipitation deficit characteristics and circulation patterns varies. Therefore, process-split rules*

*according to dry/wet spells in sect. 4 are designed to assign drought process segments to different dry/wet spells. Meanwhile, gridded values in the fields of 200 hPa/500 hPa HGT and SST are transformed into gridded values of Standardized Anomalies (SA) in sect. 5. Basically, maps of atmospheric/oceanic SA during drought process segments within the same dry/wet spells are the important inputs of predictor construction. After Empirical Orthogonal Function (EOF) analysis are conducted on these SA-based maps, the first leading EOF modes are used to build up predictors (sect. 5). Further, synchronous statistical*

*relationship between SA-based predictors and SPI3 are calibrated with the method of stepwise regression in sect. 6. The National Centers for Environmental Prediction / National Center for Atmospheric Research (NCEP/NCAR) Reanalysis datasets and the NCEP Climate Forecast System Version 2 (CFSv2) operationally forecasted datasets are used to force the synchronous statistical relationship, respectively. Simulated and predicted 90-day prospective SPI3 time series are output of sect. 7. With the help of angle-based rules of drought outlook, simulated and predicted SPI3 time series are transformed to*

*five kinds of drought outlook, which are easily accessible to end water managers.*"

[Figure]

Figure 1. Brief introduction about sequential procedures of the drought prediction model construction

Accordingly, when it comes to specified sections, we will illustrate methodology and results as follows:

Additionally, subsections 3.4 and 3.5, which bring little to the original version, will be simplified and illustrated in section 5.1 and section 6.1 in the potentially revision paper.

We think the potentially revised version will be improved a lot and easy for readers to follow and process it.

\*Major comments and general questions\*

- Introduction: Even if it becomes clear early in the paper, I think it should be stated that the droughts studied are restricted to meteorological droughts.

RESPONSE:

We will add illustration about drought types in the first paragraph of the Introduction section, which are as follows: "In the present study, drought prediction is restricted to meteorological drought, which is associated with long-term precipitation deficit."

- Section Methods: I was missing descriptions of the computation of the SPI, the EOF analysis, as well as of the step-wise regression used to build the prediction model. These could simply be described in very brief sentences.

RESPONSE:

Thank you for this comment. Actually, lacking the description of SPI3 computation is also pointed out by Referee#1. In the potentially revised version, we will add brief but important description about the computation of the SPI, the EOF analysis and the step-wise regression where necessary. Relevant main description designed for the revised version is as follows.

(1) SPI calculation

"SPI3 was used as the drought index for seasonal drought recognition and prediction in this study, and the period for SPI3 calculation is 1979–2014. Traditionally, the SPI3 set is moving in the sense that each month a new value is determined from the previous 3 months (McKee and Kleist, 1993). To obtain seasonal drought processes at the one-day timescale, we chose to update SPI3 everyday, which was also recommended by the World Metrological Organization (2012). Compared with the traditional method, the essential difference is that the interval for SPI3 calculation has been extended from 12 months to 365

days, while the moving window has changed from one month to one day. However, no changes happen to relevant mathematic procedures. Specified illustrations and details about how to calculate SPI3 updated everyday are shown as Fig. 3."

[Figure]

**Figure 3.** Illustration of calculating SPI3 updated everyday. The letter "E" represents value existence, while the letter "N" represents no relevant data.

(2) the EOF analysis

"Empirical Orthogonal Function (EOF) analysis (Wilks, 2011) is introduced to decompose spatio-temporal dataset of drought-related atmospheric/oceanic SA into spatially stationary coefficients (leading modes) and time-varying coefficients (principal component). In the same dry/wet spell, the EOF analysis is conducted on atmospheric/oceanic SA from all severe drought process segments and all extreme drought process segments respectively. The identical work of EOF analysis is applied to all the four dry/wet spells."

(3) the step-wise regression

"To build statistical models, the method of stepwise regression is introduced. Stepwise regression (Afifi and Azen, 1972) is a method of fitting multiple regression models, in which a predictive variable is considered for addition to or subtraction from the set of explanatory variables based on statistically significant extent or loss. In the present study, it is used to build the synchronous statistical relationship between all 90-day-accumulated SA-based predictors and the prediction target SPI3."

- Lines 112-114: Could you please explain why you chose the first date of the period as the beginning for the drought period? Couldn't that lead to overestimating the duration of the droughts, and subsequently influence the selection/use of predictors?

RESPONSE:

Yes, we could explain the reason for extended drought processes. Actually, due to the timescale of SPI3,
the SPI3 value on the start date of an identified drought process actually reflects drought-inducing precipitation information 90 days before it. It also corresponds to the situation that the SPI3 value is firstly less than -0.5 and the severe drought indeed comes, which is as much as important as those during the identified drought processes. Therefore, to extract drought-related atmospheric/oceanic anomalies more comprehensively, the start date of the drought process is extended to 90 days before it, prior to the drought
process division. We think it is also necessary and is important part of extended drought processes, despite the overestimated drought duration and subsequently influence on the selection of predictors.

Originally, Lines 112-114 are unclear and easily result in misunderstanding. To make it clear and logically improved, we will rewrite relevant sentences in the potentially revised version for clarity.

- Line 142: Are these the circulation pattern variables used in the building of the model? If so, it could be worth emphasizing them throughout the Methods section when appropriate.

RESPONSE:

Yes, they are. Actually, the term "atmospheric and oceanic anomalies", which is also expressed as "large-scale circulation patterns", is specified as "200 hPa/500 hPa HGT and SST". Since these three terms
express the same meanings, we have emphasized them throughout the Methods section where appropriate.

- Lines 148-150: in my opinion, these lines state analyses that have been carried out and do not really inform on the methodology itself. A brief sentence describing the EOF analysis could be useful here. Knowing the severe and extreme drought process segments at this stage could help towards a more
pragmatic description of the method.

RESPONSE:

The descriptive text in Lines 148-150 was used to explain reasons rather than describe methods and approaches. We will remove the statement components in the potentially revised version. Besides, brief introduction and application about the EOF analysis will be also added in this section.

In terms of "Knowing the severe and extreme drought process segments at this stage", we took two measures to show this information. First, we will add a general flow diagram, in which "Knowing process segments is previous to the EOF analysis" will be expressed. Second, in the potentially revised version, sect. 4 "Drought process division according to dry/wet spells" will be prior to sect. 5 "Predictor construction". This measure followed the comment of "Theme-work steps", in which the continuity
between the steps could be easier to follow.

- Lines 162-163 (also see previous comment): The sentence "All the atmospheric and oceanic predictors from all the dry/wet spells were adequately used for model calibration, which reflected drought-related information as integrally as possible." Does not seem to be supported by anything at this stage. I would
suggest moving it to the Results section if appropriate, or reformulating the sentence.
RESPONSE:
Thank you for pointing out it. The original idea is to express that it involves drought-related information as integrally as possible, despite one simple stepwise regression equation. Actually, this sentence is not supported and seems unnecessary in this part. We will remove this sentence in the potentially revised
version. Additionally, in Lines 314-321 of Discussion section in the original version, we have also expressed it.

- Section 3.6: I would have liked the authors to explain the advantage of this method over the methods found in the literature. In addition, I think this subsection needs some clarifications.
RESPONSE:
Compared with methods of drought outlook in the literature, the method itself does not show extremely obvious or significant advantages. However, in the present study, the angle-based drought outlook is an innovative and valuable attachment products for end water managers, because it is more convenient and comprehensive compared with predicted prospective SPI3 time series.

From another aspect, the extended moving window of SPI3 calculation contributes to the application of drought outlook. In the previous studies on drought outlook, a common but distinct feature is the one-month moving window of drought indices, resulting in loss of sub-month drought information. However, in the present study, partly beneficial from the one-day moving window of SPI3, prospective 90-day SPI3 time series can be predicted. Accordingly, drought outlook can be performed. It can be updated real-time and provide more accurate discriminations about drought development. It is hard for previous methods of drought outlook to provide similar prospective drought prediction information.

Last but important, drought outlook in the original subsection needs deep clarifications. Similar comments were also made by Referee#1. In the potentially revised version, we will make considerably important changes to make it brief and clear. Despite no much revisions on the Figure and Table in the subsection, the original text has been reorganized into three paragraphs, which are namely "how to describe drought development", "general classifications of drought outlook" and "how to calculate angles and conduct angle-based drought outlook". Basically, we hope to make readers easily understand the method.

- Figure 8: could you please further detail the legend for Table 8? I believe "above table" should be changed to below. Could you describe what should be read in each column? More specifically, the column "Asses." seems to indicate when the simulation and observation agree. If this is correct, the "yes" entry for 30/6/2009 should be "-", and the "-" for 11/4/2011 should be "yes".
RESPONSE:

Yes, we can. We have followed your comments and will make relevant changes in the revised version. We will replace "above table" with "below table". We will add relevant brief illustrations about the abbreviation "Simul.", "Obs." and "Asses." in the table caption. Besides, the column "Asses." actually indicate when the simulation and observation agree, and the assessments on 30/6/2009 and 11/4/2011 will be corrected in the potentially revised version. Corresponding revision have been made in the revised version.

- Lines 287-288: Is this observation based on a visual inspection of Figure 10?

RESPONSE:

Yes, this is. The original description is not rigorous indeed. In the potentially revised version, it will be described in a more rigorous approach as follows: *"As shown in Fig. 10 (b), predicted curves performed worse than the simulated curves near the peak of the 2011 East China drought, since the prospective observation tendency is rising rather than decreasing. However, in the other three droughts, the predicted curves can indicate the drought development to different degree, and they resemble the simulated results quite well. For example, operationally reforecast curves can indicate phases of occurrence, persistence, and relief during the 2009/2010 drought in Southwest China (Fig. 10 (a))."* However, only the visual inspection is not enough. In addition to this qualitative comparison, quantitative comparison of drought outlook will be shown in sect. 8 "Drought outlook" of the potentially revised version. Additionally, comparison of predicted, simulated and observed SPI3 curves with the evolution of predicted prospective periods was shown in the third issue of Discussion section.

- Tables 8 and 9: It seems that the prediction model performs better when forecasting the 2009/2010 drought in Southwest China than in simulating it. Why do you think this happens?

RESPONSE:

We think it lies in unbelievable uncertainties despite slightly better model performance. For example, the prediction model performs worse when forecasting the 2014 North China drought than in simulating it in the original table 8 and 9. The essential difference between simulated and predicted results is forced by reanalysis data or operationally forecast data. The results based on reanalysis data is the upper limitation of the latter one. Even if forecasted results sometimes perform better, it is connected with uncertainties.

*Minor comments*

- Throughout the paper, citations were sometimes organized based on alphabetical order and sometimes based on year of publication. These should be consistent.

RESPONSE:

Thank you for pointing out this problem. We have made them uniform on the basis of alphabetical order (first) and year ascending order (secondary). For example, "(Yoon et al., 2012;Mo and Lyon, 2015;Dutra et al., 2013;Dutra et al., 2014)", which is the citation in Lines 33-34 of the original version, has been adjusted into "(Dutra et al., 2013;Dutra et al., 2014;Mo and Lyon, 2015;Yoon et al., 2012)".

    - L.32: The full name of SPI is "Standardized Precipitation Index".
    RESPONSE:
    We will replace the previous term with "Standardized Precipitation Index".

- L.69: Please explain the abbreviation "SA", as it has not been explained before in the text (only in the abstract).
    RESPONSE:
    We will add the full name "Standardized Anomalies" as a brief explanation in this position.

- Section 3 Methods: I would recommend changing the titles of subsections 3.1 to 3.6. The titles should reflect what is presented in the sections, i.e. here methods and techniques, and therefore should avoid action verbs (using, divide, apply,…). In my opinion, action verbs can be misleading and can make the reader expect results.
    RESPONSE:
Thank you for pointing out these inappropriate expression. We have followed your comments to avoid action verbs and made description clear and simple. The sub sections designed for the potentially revised version are as follows:

    **3 Identification of drought processes**
3.1 Three-month SPI updated everyday
        3.2 Drought process identification and grade classification
    **4 Drought process division according to dry/wet spells**

**5 Predictor construction**

 5.1 Atmospheric and oceanic standardized anomalies

 5.2 The first EOF leading modes of SA

 5.3 Pattern-based predictor construction

**6 Model calibration**

 6.1 Synchronous statistical relationship

 6.2 Rolling calibration year by year

**7 Drought process simulation and prediction**

 7.1 Model forcing

 7.2 Drought processes simulated by the NCEP/NCAR reanalysis datasets

 7.3 Drought Processes predicted by the CFSv2 forecast datasets

**8 Drought outlook**

 8.1 Angle-based rules

 8.2 Simulated and predicted results

- Lines 147 and 303: "spatial-temporal" and "spatio-temporal" are used in these two sentences.

RESPONSE:

We will replaced the term "spatial-temporal" with the term "spatio-temporal" in the revised version.

---

## Author Response (AR1)

**Response to comments of Anonymous Referees**

Our responses to these two referees' comments are shown below in **BLUE**, with the referees' comments shown as **BLACK** text. The relevant parts in the revised version are highlighted with **YELLOW**.

5  **Response to comments of Anonymous Referee #1**

**Overall assessments:**

The manuscript illustrates a prediction model of seasonal droughts based on atmospheric/oceanic standard anomalies (SA). In particular, the model is based on synchronous relationship between SPI3 and 90-day accumulated SA anomalies.

10  Although the paper addresses an interesting topic within the scope of the journal, by proposing a novel methodology, I believe it cannot be published in its current form. My main criticisms are related to the fact that the proposed methods are poorly described or are unclear in several parts of the manuscript.

RESPONSE:

Thank you for your feedback about this manuscript. Actually, the synchronous predictor-SPI3 statistical
15  relationship forced by dynamical products, together with process prediction, are new and valuable attempt in the field of drought prediction. Besides, the process prediction model performs well at predicting seasonal drought development, despite its weakness in predicting drought severity. It is also an important result. As a whole, the paper actually addresses an important topic with a novel methodology.

Since it is a complete drought process prediction model, the procedure of model construction contains
20  adequate but necessary information. Although we tried our best to illustrate it, the original manuscript still lack clarity. With comments you and Referee#2 made, we have realized the problems to solve. Large amounts of work have been conducted to improve it, especially in the structure of the manuscript. In the revised version, we will give up the expression pattern of methodology and result section. Instead, we will choose the "theme-workstep" pattern for clarity, which is the comment from Referee #2. By doing
25  so, the continuity between the steps could be easier to follow. For example, a flow diagram map of model construction will be inserted in the end of the Introduction section. Accordingly, a brief but general introduction about the sequential procedures will also be given. Sections and sub-sections will be adjusted,

following the sequential procedures of model construction. Additionally, brief but necessary text description, tables and figures will be added in the feasible position. We hope the quality of the manuscript will be improved as much as possible, and it can be more readable and easily understood.

**Major comments:**

**- in Section 3.1,**

details on SPI computation (which seems to be different from the approach originally proposed by McKee et al., 1993) are lacking;

RESPONSE:

Thank you for pointing out this problem. We have added a flow chart to illustrate the steps of calculating SPI3 updated daily in detail. Besides, we have also made text description clear and simple. The revised text description and flow chart (**Fig. 3** in the revised version) are shown in **Line 115-124** of the revised version. Additionally, **Fig. 4** in the revised version can partly indicate the daily calculation of SPI3.

**- in Section 3.2,**

division of drought processes is rather obscure. Why do you need to split years in dry/wet periods? SPI is computed on a 90-day period, but some of the identified spells (see table 2) cover a shorter period. How do you deal with this issue? What do you mean with initial-segment days (see lines 125-129)? Figure 3 is unintelligible.

RESPONSE:

Thank for your valuable and advisable feedback, which help me realize the problems and make the description clearer. Corresponding responses are organized as follows:

(1) Why to split years into dry/wet periods

RESPONSE: Essentially, it serves the following step of predictor construction, in which drought-related atmospheric/oceanic anomalies within the same dry/wet spells are extracted and used for anomaly-based predictor construction. The main reason is that drought-related circulation patterns during different dry/wet periods are different. As illustrated in Lines 108-111 in the original version, one complete drought process usually goes through one or several dry/wet spells. Different dry/wet spells usually correspond to

various precipitation deficit characteristics and circulation patterns. Therefore, it is appropriate to divide drought processes into different segments and assign these segments into different dry/wet spells.

(2) "SPI is computed on a 90-day period, but some of the identified spells (see table 2) cover a shorter period."

RESPONSE: Actually, connections among timescale of SPI3, drought processes and dry/wet spells need to be illustrated indeed. SPI is computed on a 90-day period (SPI3), used to identify seasonal (90-day timescale) drought processes. Dry/wet spells are used to split identified complete drought processes. However, timescale of SPI3 and dry/wet spells have no relationship with each other. We think that the cause of confusion lies in the originally implicit description about SPI3 calculation and its application in seasonal drought process identification. In the revised version, the explicit description and two feasible sketch maps have been provided in Line 115-124 of the revised version.

(3) the expression of initial-segment days

RESPONSE: Initial segments are actually the split drought process segments according to dry/wet spells, which are used to compute Intersection Proportion (IP). The previous description about these two terms are confusing. In the revised paper, we have replaced "Herein, IP is the proportion of initial-segment days in days of involved spells." with the new expression "*Herein, IP is the proportion of initial segments accounting for relevant dry/wet spells, and the initial segments (e.g., D1, D3 and D4 in Fig. 6) refer to parts of one drought process split with dry/wet spells. As shown in Fig. 6, one complete process is first transformed into several initial segments according to dry/wet spells*" (Line 165-167 of the revised version).

(4) Figure 3 is unintelligible

RESPONSE: The original expression is implicit and unintelligible indeed. We think two places need to be revised. On one hand, it lacks calculation expression of IP[0] and IP[-1]. In the revised version, we have take the processes through two dry/wet spells as an example, and add the simple information of IP[0] and IP[-1] calculation. On the other hand, the original sketch maps about processes through more than two dry/wet spells lack the information before assignment. In the revised version, this problem have been solved. To sum up, necessary labels have been added in Fig. 6 of the revised version (Corresponding to Fig. 3 in the original version) for clarity.

**- in Section 3.6,**

the description of the angle comparison approach is rather messy. Please clarify and check the correctness of mathematical notations (i.e. subscripts of the angles). What is R2 in Table 3 and how is it calculated?

RESPONSE:

Actually, the implicit description of the angle comparison approach is also pointed out in the comments Referee #2 made. In the revised version, we have transformed the original text into three sub issues. They are namely "how to describe drought development", "general classifications of drought outlook" and "how to calculate angles and conduct angle-based drought outlook". By doing so, we hope that it can be explicit and easily understood. Relevant contents can be found in **Line 304-327** of the revised version.

We have also checked mathematical notations (i.e. subscripts of the angles). Currently, we have found two errors. One is a mathematical notation in the caption of original Table 3 (Corresponding to **Table 8** in the revised version). We have changed "$\alpha_i$ is greater than critical angle $\alpha_{2i}$" into "$\alpha_i$ is greater than critical angle $\alpha_{3i}$" (**Line 330** in the revised version). The other is the reversal of "$\alpha_i$" and "$\alpha_{1i}$" in the original fig. 5 (b). It has been corrected in the revised version. Besides, the original description about mathematical notations is confusing indeed. In the revised version, it has been clarified in the sub issue of "how to calculate angles and conduct angle-based drought outlook".

Finally, R2 represents the ratio of specific days in the period of the predicted prospective 46–90 days. These specific days meet the criteria that $\alpha_i$ is greater than critical angle $\alpha_{3i}$. It is dividing selected specific days by 45 (the $46^{th}$ - $90^{th}$ day) days, which can be found in the definition of R2. Illustration about R2 calculation can be found in **Line 329-330** of the revised version.

**- in Section 4.1,**

please add further information on the content of Table 5.

[Supplementary comment: I guess that Table 5 is coherent with Table 1 (drought classification) and Table 2 (division of annual period). What is unclear to me is the division of drought process (initial segment days?) illustrated at lines 125-129 and in Figure 3 and its application.]

RESPONSE:

Thank you for this feedback. Actually, results in original Table 5 follow the division rules in original Figure 3 and the dry/wet spells in original Table 2. However, it is incoherent with original Table 1. (Table 1 is used to assign dry/wet grades to every daily SPI3 value of an identified complete drought process. It will be further illustrated in the potentially major version.)

115    We have to admit that the original division rules itself (original Figure 3) are actually incomplete. In the revision version, we have added a new figure (**Figure 7** in the revised version) as an intermediate result of drought process division. Additionally, detailed but necessary labels will be inserted into the feasible positions in original Figure 3 (Corresponding to **Fig. 6** in the revised version), which help people understand the process division rules.

120

**- in Section 4.3,**

the model calibration procedure is also ambiguous What is F in Table 7? Please provide a list of the initial 43 predictors and the selected ones.

125    RESPONSE:

Actually, in original Table 7, F represents values of the constructed F-test statistic in the final stepwise regression, while $F_{\alpha=0.05}$ refer to F-test critical values at 5% significance level. Considering F and $F_{\alpha=0.05}$ are included in the details of procedures, we tend to delete these two columns of F and $F_{\alpha=0.05}$ in original Table 7. The new table corresponds to **Table 6** in the revised version.

130    Besides, a list of the initial 43 predictors and the selected ones have also been shown as **Table 7** in the revised version. Since all the initial 43 predictors are shown, we have also added the information of predictor construction in the 200 hPa HGT field into the manuscript (**Table 5** in the revised version). Accordingly, the first leading EOF modes of SA for 200 hPa HGT have been also shown as **Fig. 9** in the revised version.

135

**- in Section 4.4,**

the synchronous stepwise-regression relationship should be described in-depth.

[I suggest to better clarify:

- the structure of the multiple regression models (linear or not?);

140 - the explained variables

(the first PCs of SA predictors reported in Table 6?) and which criterion is used

to select the most significant ones;

- how the calibration and validation periods have been chosen (see Table 7) and which of them is finally

applied.]

145 RESPONSE:

Thank you for pointing out this problem about clarity. Stepwise regression is also multiple and linear. Essentially, stepwise regression selects a best subset of explanatory variables used for model construction while all the explanatory variables are used to build multiple linear regression model. Additionally, positive and negative pattern areas on the first leading EOF modes are used to build SA predictors in our

150 study. SA predictors reported in original Table 6 is actually explanatory variables, instead of the first PCs (Principal Component) of SA predictors.

The general description about criterion used to select the most significant ones (Afifi and Azen, 1972) is as follows. "The technique of stepwise regression which selects a best subset according to the following procedure: The first step selects the single variable which best predicts Y. The second step finds the

155 variable which best predicts Y given the first variable entered. In the steps that follow, either: (a) a variable is entered which best improved the prediction of Y given all the variables entered from the previous steps; or (b) a variable is removed from the set of predictors if its predictive ability falls below a given level. The process is terminated when no further variable improves the prediction of Y".

However, these aforementioned information is detailed. Considering large amounts of information this

160 manuscript contains, we tend to give a brief but important introduction about the structure, explanatory variables and criterion of stepwise regression in the revised version (**Line 238-240** in the revised version). Additionally, the citation of the introduction about stepwise regression (Afifi and Azen, 1972) will be also provided.

References:

165 Afifi, A. A., and Azen, S. P.: Statistical analysis: a computer oriented approach, Academic press, 1972.

In terms of selection approaches of calibration and validation periods, two reasons are responsible for it. First, we want to examine model performance on the 2009/2010 Southwest China drought. Therefore, the end of the first calibration period is 31 December 2008. Second, consistent with the drought prediction year by year, the calibration period is running and extended to the day before the new year. For example, the seasonal drought prediction model calibrated from 1 Jan 1983 to 31 Dec 2011, is used for everyday initial prediction time in the whole year of 2012. When it comes to every initial drought prediction in the year of 2013, the corresponding drought model is calibrated from 1 Jan 1983 to 31 Dec 2012. Accordingly, every SPI3 value in original Fig.9 and original Fig.10 is simulated or predicted, using the drought prediction model with corresponding calibration period. We think the original description in original section 4.3 is unclear and confusing. We have make a brief but necessary explanation about the selection of calibration in **Line 245-254** of the revised version.

Overall, the lack of clarity in the methodology makes difficult to verify the quality of the derived results. RESPONSE:

Thank you for pointing out the lack of clarity. Actually, large amounts of revision work is being conducted to improve it. For example, a general flow diagram of model construction have been provided at the end part of the Introduction section, aiming at a brief instruction about sub-sections. Besides, **Fig. 6** in the revised version (Corresponding to the original Figure 3) has become readable, by means of adding some necessary labels. As an important revision designed for revised version, three measures of **general introduction to the sequential procedures of model construction** is as follows:

**(1) Descriptive texts in the end of the Introduction section (Line 75-90 in the revised version)**

 **(2) General flow chart (Fig. 1 in the revised version)**

 **(3) Section assignments**

3 Identification of drought processes

    3.1 Three-month SPI updated daily

    3.2 Drought process identification and grade classification

4 Drought process division according to dry/wet spells

Finally, I would also suggest the authors to revise the language of the manuscript in order to make it more fluid and comprehensible.

RESPONSE:

Thank you for this comment. We have followed almost all the comments, and this manuscript had been revised as much as possible by ourselves. Subsequently, we have further invited professional editors at Editage, a division of Cactus Communications to revise it and improve the language quality (). We try our best to make this manuscript more fluid, readable and comprehensible.

**CERTIFICATE OF**
**ENGLISH EDITING**

This document certifies that the paper listed below has been edited to ensure that the language is clear and free of errors. The edit was performed by professional editors at Editage, a division of Cactus Communications.The intent of the author's message was not altered in any way during the editing process. The quality of the edit has been guaranteed, with the assumption that our suggested changes have been accepted and have not been further altered without the knowledge of our editors.

**TITLE OF THE PAPER**

A conceptual prediction model for seasonal drought processes using atmospheric and oceanic Standardized Anomalies and its application to four recent severe drought events in China

**AUTHORS**

Zhenchen Liu

**JOB CODE**

YXENC_2_3

[Figure]

Signature

[Figure]

Vikas Narang,
Vice President, Author Services, Editage

Date of Issue
**May 31, 2017**

Editage, a brand of Cactus Communications, offers professional English language editing and publication support services to authors engaged in over 500 areas of research. Through its community of experienced editors, which includes doctors, engineers, published scientists, and researchers with peer review experience, Editage has successfully helped authors get published in internationally reputed journals. Authors who work with Editage are guaranteed excellent language quality and timely delivery.

[Figure]

**Contact Editage**

Worldwide
request@editage.com
+1 877-334-8243
www.editage.com

Japan
submissions@editage.com
+81 03-6868-3348
www.editage.jp

Korea
submit-korea@editage.com
1544-9241
www.editage.co.kr

China
fabiao@editage.cn
400-005-6055
www.editage.cn

Brazil
inquiry.brazil@editage.com
0800-892-20-97
www.editage.com.br

Taiwan
submitjobs@editage.com
02 2657 0306
www.editage.com.tw

**Response to comments of Anonymous Referee #2**

This paper proposes a statistical drought prediction model based on atmospheric and oceanic variables. The authors first identify severe and extreme drought events based on the SPI3 and identify predictors for these events. Based on these, they build a drought prediction model and propose a drought outlook. The performance of the full chain is then illustrated in the case of four drought events in China.

*General comment*

I believe that this paper is a valuable contribution to the special issue. However, I believe that, in its current form, it is hard for the reader to follow and process the large amount of information it contains. For clarification, I would suggest reorganizing the paper. Indeed, some of the subsections in the Methods section bring little to the paper in their current state (especially subsections 3.4 and 3.5). I could suggest two ways (non-restrictive) to reorganize the Methods and Results sections. (1) The first suggestion would be to keep the current structure but making sure that the Methods section (a) is more detailed and explains even briefly all methods, including the computation of the SPI, the step-wise regression and the EOF analysis, and (b) excludes statements on what has been done (move to the Results section). (2) The second way could be to separate the paper by "themes" or "work steps" as listed at the end of the introduction: this way, the continuity between the steps could be easier to follow, and, for instance, the drought periods and predictors would be available to the reader to understand the steps of "structuring predictors" and "building the prediction model".

RESPONSE:

Thanks for your admiration about the scientific values of this manuscript. Actually, it is a new and valuable attempt of seasonal drought process prediction, which hardly appear in the previous study.

The most important issue to solve is the lack of clarification, especially in the Methods section and Results section. In the revised version, we tend to choose the second way you recommended, which separate the paper by "themes" or "work steps".

To achieve it, we have added a flow diagram map of model construction (**Fig. 1** in the revised version) at the end of the Introduction section and give a brief introduction about the sequential procedures (**Line 75-90** in the revised version).

Accordingly, specified sections are as follows:

*3 Identification of drought processes*

*3.1 Three-month SPI updated everyday*

*3.2 Drought process identification and grade classification*

*4 Drought process division according to dry/wet spells*

*5 Predictor construction*

*5.1 Atmospheric and oceanic standardized anomalies*

*5.2 The first EOF leading modes of SA*

*5.3 Pattern-based predictor construction*

*6 Model calibration*

*6.1 Synchronous statistical relationship*

*6.2 Rolling calibration year by year*

*7 Drought process simulation and prediction*

*7.1 Model forcing*

*7.2 Drought processes simulated with the NCEP/NCAR reanalysis datasets*

*7.3 Drought Processes predicted with the CFSv2 forecast datasets*

*8 Drought outlook*

*8.1 Angle-based rules*

*8.2 Simulated and predicted results*

Additionally, the original subsections 3.4 and 3.5, which bring little to the original version, have been simplified and illustrated in section 5.1 and section 6.1 in the revision paper.

Overall, we think the revised version will be improved a lot and easy for readers to follow and process it.

\*Major comments and general questions\*

- Introduction: Even if it becomes clear early in the paper, I think it should be stated that the droughts studied are restricted to meteorological droughts.

RESPONSE:

We have added illustration about drought types in the first paragraph of the Introduction section (**Line 30-31** in the revised version), which are as follows: "In the present study, drought prediction is restricted to meteorological drought, which is associated with long-term precipitation deficit."

- Section Methods: I was missing descriptions of the computation of the SPI, the EOF analysis, as well as of the step-wise regression used to build the prediction model. These could simply be described in very brief sentences.

RESPONSE:

Thank you for this comment. Actually, lacking the description of SPI3 computation is also pointed out by Referee #1. In the revised version, we have added brief but important description about the computation of the SPI, the EOF analysis and the step-wise regression where necessary. Relevant main description in the revised version is as follows.

 (1) The brief description of SPI3 calculation can be found in **Line 115-121** and **Fig. 3** of the revised version.

 (2) The method of the EOF analysis can be found in **Line 196-198** of the revised version.

 (3) The method of the step-wise regression can be found in **Line 238-240** of the revised version.

- Lines 112-114: Could you please explain why you chose the first date of the period as the beginning for the drought period? Couldn't that lead to overestimating the duration of the droughts, and subsequently influence the selection/use of predictors?

RESPONSE:

Yes, we could explain the reason for extended drought processes. Actually, due to the timescale of SPI3, the SPI3 value on the start date of an identified drought process actually reflects drought-inducing precipitation information 90 days before it. It also corresponds to the situation that the SPI3 value is firstly less than -0.5 and the severe drought indeed comes, which is as much as important as those during the

300 identified drought processes. Therefore, to extract drought-related atmospheric/oceanic anomalies more comprehensively, the start date of the drought process is extended to 90 days before it, prior to the drought process division. We think it is also necessary and is important part of extended drought processes, despite the overestimated drought duration and subsequently influence on the selection of predictors.

Originally, Lines 112-114 are unclear and easily result in misunderstanding. To make it clear and logically
305 improved, we have rewritten relevant sentences (**Line 149-153** in the revised version) for clarity.

- Line 142: Are these the circulation pattern variables used in the building of the model? If so, it could be worth emphasizing them throughout the Methods section when appropriate.

RESPONSE:
310 Yes, they are. Actually, the term "atmospheric and oceanic anomalies", which is also expressed as "large-scale circulation patterns", is specified as "200 hPa/500 hPa HGT and SST". Since these three terms express the same meanings, we have emphasized them throughout the Methods section where appropriate.

- Lines 148-150: in my opinion, these lines state analyses that have been carried out and do not really
315 inform on the methodology itself. A brief sentence describing the EOF analysis could be useful here. Knowing the severe and extreme drought process segments at this stage could help towards a more pragmatic description of the method.

RESPONSE:
The descriptive text in Lines 148-150 was used to explain reasons rather than describe methods and
320 approaches. We have removed the statement components in the revised version. Besides, brief introduction and application about the EOF analysis have been also added in **Line 196-198** of the revised version.

In terms of "Knowing the severe and extreme drought process segments at this stage", we took two measures to show this information. First, we have added a general flow diagram (**Fig. 1** in the revised
325 version), in which "Knowing process segments is previous to the EOF analysis" have been expressed. Second, in the revised version, **Sect. 4** "Drought process division according to dry/wet spells" will be

prior to **Sect. 5** "Predictor construction". This measure followed the comment of "Theme-work steps", in which the continuity between the steps could be easier to follow.

330 - Lines 162-163 (also see previous comment): The sentence "All the atmospheric and oceanic predictors from all the dry/wet spells were adequately used for model calibration, which reflected drought-related information as integrally as possible." Does not seem to be supported by anything at this stage. I would suggest moving it to the Results section if appropriate, or reformulating the sentence.

RESPONSE:

335 Thank you for pointing out it. The original idea is to express that it involves drought-related information as integrally as possible, despite one simple stepwise regression equation. Actually, this sentence is not supported and seems unnecessary in this part. We have removed this sentence in the revised version. Additionally, in **Line 366-372** of the revised version, we have also expressed it.

340 - Section 3.6: I would have liked the authors to explain the advantage of this method over the methods found in the literature. In addition, I think this subsection needs some clarifications.

RESPONSE:

Compared with methods of drought outlook in the literature, the method itself does not show extremely obvious or significant advantages. However, in the present study, the angle-based drought outlook is an

345 innovative and valuable attachment product for water resource managers, because it is more convenient and comprehensive compared with predicted prospective SPI3 time series.

From another aspect, the extended moving window of SPI3 calculation contributes to the application of drought outlook. In the previous studies on drought outlook, a common but distinct feature is the one-month moving window of drought indices, resulting in loss of sub-month drought information. However,

350 in the present study, partly beneficial from the one-day moving window of SPI3, prospective 90-day SPI3 time series can be predicted. Accordingly, drought outlook can be performed. It can be updated real-time and provide more accurate discriminations about drought development. It is hard for previous methods of drought outlook to provide similar prospective drought prediction information.

Last but important, drought outlook in the original subsection needs deep clarifications. Similar comments were also made by Referee #1. In the revised version, we have made considerably important changes to make it brief and clear. Despite no much revisions on the original Figure 5 and Table 3 in the subsection (Corresponding to Fig. 14 and Table 8 of the revised version, respectively), the original text has been reorganized into three paragraphs, which are namely "how to describe drought development", "general classifications of drought outlook" and "how to calculate angles and conduct angle-based drought outlook" (Line 304-327 in the revised version). By doing so, we hope to make readers easily understand the method.

- Figure 8: could you please further detail the legend for Table 8? I believe "above table" should be changed to below. Could you describe what should be read in each column? More specifically, the column "Asses." seems to indicate when the simulation and observation agree. If this is correct, the "yes" entry for 30/6/2009 should be "-", and the "-" for 11/4/2011 should be "yes".

RESPONSE:

Yes, we can. We have followed your comments and made relevant changes in the revised version. We have also replaced "above table" with "below table". We have added relevant brief illustrations about the abbreviation "Simul.", "Obs." and "Asses." in the table caption. Besides, the column "Asses." actually indicate when the simulation and observation agree, and the assessments on 30/6/2009 and 11/4/2011 have been corrected in the potentially revised version. Corresponding revision can be found in Table 9 of the revised version.

- Lines 287-288: Is this observation based on a visual inspection of Figure 10?

RESPONSE:

Yes, this is. The original description is not rigorous indeed. In the revised version, it has been described in a more rigorous approach (see Line 291-295 in the revised version).

However, only the visual inspection is not enough. In addition to this qualitative comparison, quantitative comparison of drought outlook has been shown in Sect. 8 "Drought outlook" of the revised version. Additionally, comparison of predicted, simulated and observed SPI3 curves with the evolution

of predicted prospective periods are also shown in the third issue of Discussion section (**Line 373-386** in the revised version).

385   - Tables 8 and 9: It seems that the prediction model performs better when forecasting the 2009/2010 drought in Southwest China than in simulating it. Why do you think this happens?

RESPONSE:

We think it lies in uncertainties of forecast errors. The difference between simulated and predicted results is forced by reanalysis data or operationally forecast data. Essentially, the results based on

390   reanalysis data are more accurate than those based on operationally forecast products. However, due to uncertainties of forecast errors, forecasted results may perform better or worse than simulated ones, or resemble the simulated ones. All of these three situations are possible to occur. For example, the prediction model performs worse when forecasting the 2014 North China drought than in simulating it in the original table 8 and 9 (**Table 9 and Table 10** in the revised version).

395   Additionally, there may exist another possibility that uncertainties of synchronous statistical relationship and forecast errors are mutually eliminated. The synchronous statistical relationship can not describe the essential predictor-SPI3 connection absolutely, which surely bring errors. When it came across errors of dynamical products, errors of the "whole" model (synchronous statistical relationship + dynamical force) may be eliminated.

400

  *Minor comments*

  - Throughout the paper, citations were sometimes organized based on alphabetical order and sometimes based on year of publication. These should be consistent.

405   RESPONSE:

Thank you for pointing out this problem. We have made them uniform on the basis of alphabetical order (first) and year ascending order (secondary). For example, "(Yoon et al., 2012;Mo and Lyon, 2015;Dutra et al., 2013;Dutra et al., 2014)", which is the citation in Lines 33-34 of the original version, has been adjusted into "(Dutra et al., 2013;Dutra et al., 2014;Mo and Lyon, 2015;Yoon et al., 2012)".

410

- L.32: The full name of SPI is "Standardized Precipitation Index".

RESPONSE:

We have replaced the previous term with "Standardized Precipitation Index".

415

- L.69: Please explain the abbreviation "SA", as it has not been explained before in the text (only in the abstract).

RESPONSE:

We have added the full name "Standardized Anomalies" as a brief explanation in this position.

420

- Section 3 Methods: I would recommend changing the titles of subsections 3.1 to 3.6. The titles should reflect what is presented in the sections, i.e. here methods and techniques, and therefore should avoid action verbs (using, divide, apply,…). In my opinion, action verbs can be misleading and can make the reader expect results.

425 RESPONSE:

Thank you for pointing out these inappropriate expression. We have followed your comments to avoid action verbs and made description clear and simple. The sub sections designed for the revised version are as follows:

*3 Identification of drought processes*

430    *3.1 Three-month SPI updated everyday*

    *3.2 Drought process identification and grade classification*

*4 Drought process division according to dry/wet spells*

*5 Predictor construction*

    *5.1 Atmospheric and oceanic standardized anomalies*

435    *5.2 The first EOF leading modes of SA*

    *5.3 Pattern-based predictor construction*

*6 Model calibration*

- Lines 147 and 303: "spatial-temporal" and "spatio-temporal" are used in these two sentences.

RESPONSE:

450   We have replaced the term "spatial-temporal" with the term "spatio-temporal" in the revised version.

[revised manuscript text omitted]


[revised manuscript text omitted]

---

## Editor Decision (ED1)

Dear Authors

I really appreciate your efforts to improve the quality of your manuscript. Nonetheless, some parts are still not very clear to me.

For instance, I still do not understand the procedure applied for computing SPI3.
Following your description, a running window of 90 days is applied to daily precipitation, so that the time unit is 1-day rather than 1-month, as in the original procedure developed by McKee et al. (1993). What is doubtful is that according to Fig.3, SPI series are, apparently, computed by first fitting a gamma function to the precipitation data from the same day within the running window, for all the years included in the historical record. The gamma distribution is then transformed into a normal distribution.
In the original procedure, the precipitation totals from the same 3-month period are fitted to a probability distribution (usually gamma) and then transformed to a normal distribution, so that the SPI is referred to the last month of the considered 3-month period. In this way, SPI is designed to quantify the precipitation deficit for a 3-month timescales.
In Fig. 3, you refer to 90-day mean value. Thus, the reader is left to suppose that in SPI3 is derived by averaging daily SPI within a 90-day running period. However, this is denied in Section 3.2 and in Fig. 4, where it seems that the whole time evolution of daily SPI is considered instead. If this is correct, I suggest to improve Fig. 3 and to better clarify in Section 3.1.
Besides, following Section 3.2, the highest negative SPI3-grade is assigned to a drought process (i.e. when SPI3 values are below -0.50 for more than 30 consecutive days) when the corresponding duration is greater than 35% of the total drought duration. What is the rationale behind the choice of 35% threshold? Have you carried out some kind of sensitivity analysis on the threshold value?

With reference to Section 4 "Drought process division" and Section 5 "Predictor construction", I evaluate the revised manuscript definitively improved. My only request of clarification is with respect to the EOF analysis applied to the atmospheric and oceanic standardized anomalies. As far as I have understood from the revised manuscript, EOF analysis is applied to SA daily grid point data of 200 hPa/500 hPa HGT and SST corresponding to the drought segments with the same dry/wet spell and drought grade (extreme or severe) according to Table 4. Therefore, μ and σ are the daily grid-point mean value and daily grid-point standard deviation computed with reference to the period indicated in Table 4 (e.g. for wet spell and extreme dry condition, μ and σ are computed on daily data observed during 21/6/1997–10/9/1997 and 4/8/1998–10/9/1998). Is it correct?

With respect to Section 6.1, synchronous statistical relationship are determined between all 90-day-accumulated SA-based predictors and prediction target SPI3 by using a stepwise regression. From lines 637 "the predictor is area-averaged over all gridded SA-based variables in selected areas, such as A and B …", I understand that regression (see Table 6 and Table 7) are computed only with respect to the most significant positive and negative pattern areas in the first EOF, as reported in Figs. 8, 9 and 10 and Table 5. If I am right, the prediction model, based on the considered predictors, is performed only with reference to limited areas, rather than on the whole China, as Fig. 11 and Fig. 12 imply. Would you please clarify this issue?

Equation at line 360 (page 7) is maybe out of position.

There are still some typos in the text, such as "metrological" or "El nirio". Please check!

---

## Author Response (AR2)

**Response to comments of Anonymous Referees**

Our responses to these two referees' comments are shown below in **BLUE**, with the referees' comments shown as **BLACK** text.

**Response to comments of Anonymous Referee #1**

**Overall assessments:**

Dear Authors

I really appreciate your efforts to improve the quality of your manuscript. Nonetheless, some parts are still not very clear to me.

RESPONSE:

Thank you for your appreciation on the quality improvement, and we are so grateful for these comments below. These comments, such as the procedure of SPI3 computation and computing $\mu$ and $\sigma$, really help me realize that the revised expression is still information-missing in some certain places. If not improved or corrected in time, they tend to be confusing and difficult to understand for potential readers. The detailed responses to these comments are listed below and relevant contents have been improved in the latest revised manuscript.

**Major comments:**

For instance, I still do not understand the procedure applied for computing SPI3.

Following your description, a running window of 90 days is applied to daily precipitation, so that the time unit is 1-day rather than 1-month, as in the original procedure developed by McKee et al. (1993). What is doubtful is that according to Fig.3, SPI series are, apparently, computed by first fitting a gamma function to the precipitation data from the same day within the running window, for all the years included in the historical record. The gamma distribution is then transformed into a normal distribution. In the original procedure, the precipitation totals from the same 3-month period are fitted to a probability distribution (usually gamma) and then transformed to a normal distribution, so that the SPI is referred to the last month of the considered 3-month period. In this way, SPI is designed to quantify the precipitation deficit for a 3-month timescales.

In Fig. 3, you refer to 90-day mean value. Thus, the reader is left to suppose that in SPI3 is derived by averaging daily SPI within a 90-day running period. However, this is denied in Section 3.2 and in Fig. 4, where it seems that the whole time evolution of daily SPI is considered instead. If this is correct, I suggest to improve Fig. 3 and to better clarify in Section 3.1.

RESPONSE:

Thank you for this comment very much; Otherwise, we will not realize that the original expression of Fig.3 is possible to be clear and misleading. Actually, with respect to the correct approach of SPI3 calculation, we quite agree with what you understand and the original procedure developed by McKee et al. (1993). Accordingly, the third step in the original Fig.3 is to calculate 90-day-mean precipitation data first, and then calculate SPI3 from the same day based on the computed 90-day-mean precipitation data.

For better clarity, in Fig.3 of the latest revision, "Calculating 90-day-mean precipitation

data" and "calculating SPI3 series updated daily" are shown in a sequential way, respectively. In addition, we also have replaced the original expression "For values with the same day number" with "For 90-day-mean values with the same day number" in the fourth step of the latest version.

Besides, following Section 3.2, the highest negative SPI3-grade is assigned to a drought process (i.e. when SPI3 values are below -0.50 for more than 30 consecutive days) when the corresponding duration is greater than 35% of the total drought duration. What is the rationale behind the choice of 35% threshold? Have you carried out some kind of sensitivity analysis on the threshold value?
RESPONSE:
In terms of the choice of 35% threshold, some analysis was also conducted. As shown in Figure 1 (below the text), numbers of identified severe and extreme processes generally declined with the increase of threshold percent. When the threshold percent is relatively low, a large number of severe and extreme processes with "low standard" were identified, and some of them essentially can not meet the extreme or severe criteria. However, when it comes to be no less than 45%, the figure for identified processes is relatively low and even no extreme drought processes were identified in East China and North China.
Considering these two aspects, we finally chose the 35% threshold and thought it reasonable. Due to the limitation of manuscript pages, we ignored this select procedure in the manuscript.

[Figure]

**Figure 1.** Relationship between numbers of identified severe and extreme processes and threshold percent in East, North and Southwest China.

With reference to Section 4 "Drought process division" and Section 5 "Predictor construction", I evaluate the revised manuscript definitively improved. My only request of clarification is with respect to the EOF analysis applied to the atmospheric and oceanic standardized anomalies. As far as I have understood from the revised manuscript, EOF analysis is applied to SA daily grid point data of 200 hPa/500 hPa HGT and SST corresponding to the drought segments with the same dry/wet spell and drought grade (extreme or severe) according to Table 4. Therefore, $\mu$ and $\sigma$ are the daily grid-point mean value and daily grid-point standard deviation computed with reference to the period indicated in Table 4 (e.g. for wet spell and extreme dry condition, $\mu$ and $\sigma$ are computed on daily data observed during 21/6/1997–10/9/1997 and 4/8/1998–10/9/1998). Is it correct?
RESPONSE:
No, it is not correct. Personally, some necessary information about methods of $\mu$ and $\sigma$ computation need to be specified. Without them, it may lead to being misunderstood. Therefore, we really thank you for pointing out this problem.

Actually, in original section 5.1, the original expression "The climatological periods are 1979–2008 for 200 hPa/500 hPa HGT and 1982–2008 for SST, respectively." has contained the computation methods of $\mu$ and $\sigma$. However, for more clarity, some supplement has been added into section 5.1 of the latest revision, which are as follows: "For example, with respect to one certain grid point, both the mean 1 January 500 hPa HGT value and associated standard deviation are computed on the 1 January 500 hPa HGT datasets observed during 1979–2008 at the grid point".

With respect to Section 6.1, synchronous statistical relationship are determined between all 90-day-accumulated SA-based predictors and prediction target SPI3 by using a stepwise regression. From lines 637 "the predictor is area-averaged over all gridded SA-based variables in selected areas, such as A and B …", I understand that regression (see Table 6 and Table 7) are computed only with respect to the most significant positive and negative pattern areas in the first EOF, as reported in Figs. 8, 9 and 10 and Table 5. If I am right, the prediction model, based on the considered predictors, is performed only with reference to limited areas, rather than on the whole China, as Fig. 11 and Fig. 12 imply. Would you please clarify this issue?

RESPONSE:

Yes, we would. At first, thank you for this comment. Actually, regression are computed only with respect to these significantly positive and negative areas in the first EOF, and these anomalous areas are closely related to one prediction target region, such as North China, East China or Southwest China. The original Table 6, Table 7, Figs. 8, 9 and 10 and Table 5 are all with respect to North China. North China, these figures and tables serve, has also been involved in corresponding captions. The reason is that North China was used to introduce the methods of the model construction and calibration. Due to the limitation of manuscript pages, figures and tables associated with similar procedures in East China and Southwest China are not shown in the present manuscript, but important results of process simulation and prediction are shown and analyzed in the manuscript.

Overall, process-based drought prediction model is applied in four recent severe regional drought events in China, which are droughts in East China, North China and Southwest China. Therefore, we focus on model simulation and prediction in these three study regions, but only take North China as an example of methodology introduction. To avoid misunderstanding, the three drought study regions we focus on are labelled with red boxes in Fig.2. Besides, the word "regional" has been inserted into the title of this manuscript. In addition, the last paragraph in the Introduction section has also emphasized it.

Equation at line 360 (page 7) is maybe out of position.

RESPONSE:

We have made this Equation in the appropriate position.

There are still some typos in the text, such as "metrological" or "El nirio". Please check!

RESPONSE:

We have corrected the typos in the text.

**Response to comments of Anonymous Referee #2**

**General comment**

I want to thank the authors for thoroughly taking into account the comments made by the other reviewer and myself in this revision. The changes have substantially improved the quality of the paper and now propose a clear overview of the different (and numerous) work steps presented. More specifically, I really appreciate Figures 1 and 4 as they answer many of the questions I had after reading the first manuscript.

RESPONSE:

Thank you for your appreciation about the revision. Without the advisable suggestions proposed by you and Referee #1, it is hard for us to make the manuscript clear, and easily understood.

The significant changes, which I am satisfied with, have considerably increased the length of the paper: 8 more pages in the revised version than in the original manuscript. As a general comment, I would advise authors to cut text where possible, even though I understand that it may be difficult to fit the new and relevant information into a shorter paper. Possible suggestions include: removing Figure 3, removing Figure 6 because Figure 7 and Table 4 apply in a very clear way the theoretical cases of Figure 6 to the studied drought events.

RESPONSE:

Thank you for this comment, and it is actually necessary to make it shorter. Considering the information some figures and tables expressed is helpful for potential readers to understand the procedures of model construction, we tend to move them into the Supplement part. Detailed revision and relevant reasons are as follows:

(1) We have made the original Figure 3 reserved. It explains methods of computing SPI3 updated daily, which is something new to potential readers. Referee #1 also showed its concern on it.

(2) We have made the original Figure 6 reserved, because this universal rules is valuable. Accordingly, the original Figure 7, which acts as application cases of the theoretical rules, have been moved to the Supplement part. In addition, the original Table 4 is still in its position of the manuscript.

(3) Finally, to cut text where possible, the original Figure 9 has been also moved to the Supplement part, since it is similar to the original Figure 8 but for 200 hPa HGT SA. Besides, the original Table 7 has been also moved to the Supplement part. The reason is that it contains necessary and detailed information about selected predictors and relevant coefficients, despite associated little analysis in the manuscript.

**Minor comment**

Lines 70-74: Point (3) and point (4) could be inversed to follow the new structure of the paper.

RESPONSE:

Thank you for this comment, and we have made them inversed.

[revised manuscript text omitted]
. 3–6, but similar detailed procedures in East and Southwest China were not shown in the present study. Besides, SPI3 time series during the period extending from 2009 to 2014 in North China, East China, and Southwest China were used in the process simulation. Finally, recent severe drought processes in these three regions were used to verify model performance in operational application.

105 **2 Data**

The precipitation data used were the second-version Dataset of Observed Daily Precipitation Amounts at each 0.5° × 0.5° grid point in China for 1961–2014 (http://data.cma.cn/data/detail/dataCode/SURF_CLI_CHN_PRE_DAY_GRID_0.5.html), which was kindly provided by the Climate Data Center (CDC) of the National Meteorological Information Center, China Meteorological Administration (CMA). It was initially used to calculate area-averaged precipitation in North China, East

110 China, and Southwest China (Fig. 2), which are the three Chinese drought regions investigated in this study. Atmospheric anomalies were diagnosed with respect to the NCEP/NCAR Reanalysis datasets, which has a resolution of 2.5° × 2.5° at 17 pressure levels, extending from January 1948 to the present (Kalnay et al., 1996). The National Oceanic and Atmospheric Administration (NOAA) High Resolution SST dataset, with a spatial resolution of 0.25° × 0.25° and extends from September 1981 to present (Reynolds et al., 2007), were used for SST anomaly analysis. Additionally, the NCEP Climate

115 Forecast System Version 2 (CFSv2; Saha et al., 2014) was introduced to verify operational performance of the proposed conceptual model. Since CFSv2 began on 1 April 2011, some drought events that occurred before this date were forced with the CFS reforecast output. All the reforecast and forecasted datasets are accessible on the website

120 (https://nomads.ncdc.noaa.gov/modeldata/), and 6-hourly forecasted datasets are transformed to daily timescale by a simple time-weighted mean method.

[Figure]

**Figure 2.** The geographical distribution of China's nine drought study regions (black solid curves) and provinces (light grey curves). The three regions labelled with red boxes are the focus in the present study.

125 **3 Identification of drought processes**

**3.1 Three-month SPI updated daily**

SPI3 was used as the drought index for seasonal drought recognition and prediction in this study, and the calculation period is 1979–2014. Traditionally, the SPI3 set varies with a monthly timescale; each month a new value was determined from the previous 3 months (McKee and Kleist, 1993). To obtain seasonal drought processes at the 1-day timescale, we chose to update

130 SPI3 daily, which was also recommended by the World Meteorological Organization (2012). Compared with the traditional method, the essential difference is that the interval for SPI3 calculation has been extended from 1 month to 1 day. However, no other changes relevant to mathematic procedures occur. Specified illustrations and details for calculating SPI3 updated daily are shown as Fig. 3. Prior to the detailed procedures shown in Fig.3, daily area-averaged precipitation datasets are computed first.

[revised manuscript text omitted]

$$SA = \frac{X - \mu}{\sigma}, \tag{1}$$

Where X represents daily grid-point atmospheric/oceanic circulation pattern variables, which are 200 hPa/500 hPa HGT and SST. $\mu$ and $\sigma$ are the daily grid-point mean value and daily grid-point standard deviation, respectively. The climatological periods are 1979–2008 for 200 hPa/500 hPa HGT and 1982–2008 for SST, respectively. For example, with respect to one certain grid point, both the mean 1 January 500 hPa HGT values and associated standard deviation are computed on the 1 January 500 hPa HGT datasets observed during 1979–2008 at the grid point.

[revised manuscript text omitted]

---

## Editor Decision (ED2)

**Final comments to the authors of:**

**hess-2017-136 - A conceptual prediction model of seasonal drought processes using atmospheric and oceanic Standardized Anomalies: application in four recent severe drought events in China, by Zhenchen Liu, Guihua Lu, Hai He, Zhiyong Wu, and Jian He**

Dear authors,

I appreciated the revised version and modifications, which greatly improved the manuscript. There were some significant changes, which clarified the study methodology and results, but I think some minor technical/language revision of the paper can still be done to prepare the final version of the manuscript before its publication, which I recommend (without further revision by reviewers or the Editor).

My suggestions are listed below:

- I think the title is too long, with too many adjectives. I suggest to slightly shorten it to: "A conceptual prediction model for seasonal drought processes using atmospheric and oceanic standardized anomalies: application to regional drought events in China"
- Abstract:
    - line 10: change to "we developed"
    - line 10: "based on atmospheric and oceanic…"
    - lines 11-12: "at 200 hPa and 500 hPa…"
    - line 16: change "It can make seamless drought prediction…" to "The model can make seamless drought predictions…"
    - line 17: too many adjectives (recent severe regional drought) for the world "event". Please rephrase it by checking the use of the English language.
    - lines 18-19: the last sentence ("Therefore,…") is not fully comprehensive, especially given the limitations written just before it. I suggest changing to the following: "Overall, the model can be a worthy reference for seasonal water resources management in China."
- Line 36: what do you mean by "short-range"? One day ahead? A week ahead? One month ahead? Please, specify.
- Line 39: consider changing to "…physical, complex processes…"
- Line 41: consider changing to "…input variables, a methodology and prediction…"
- Lines 44-45: it seems to me that here you are talking about hydrological drought (ESP method) instead of meteorological drought as stated earlier in lines 30/31. Could you clarify it please?
- Lines 53-54: consider changing to "Climate indices, such as… are simple, explicit and widely used. Therefore, they are…"
- Lines 65-66: consider changing to "…all these predictors, influencing different drought processes, and the 3-month SPI updated daily (hereafter, SPI3) were used to calibrate a synchronous …"
- Overall, please check when using "process" and "events". Sometimes it seems to me that you mean "drought events" and not "drought process", as, for instance, in lines 73, 77, 78, 79, 131, Fig. 4 caption, etc. If you really mean "process", please add a sentence clarifying the way you define these two different things (process and event) in the study.

- Line 76: correct to singular verb in "…a brief but general introduction with… is presented (Fig. 1), …"
- Line 79: consider changing to "Therefore, event-split rules, according to dry/wet spells, are designed to assign drought event segments to different dry/wet spells in Sec. 4"
- Line 80: consider deleting "Meanwhile,"
- Line 82: consider changing to "…are important inputs to the construction of the predictors."
- Line 86: consider changing to "…and the NCEP… operational forecast datasets are used to… relationship." Consider also deleting "respectively."
- Fig 1 caption: consider changing to "…sequential procedures described in the sections of this study for drought…"
- Line 93: consider changing to "…events in North China are used to…"
- Line 94: consider splitting into two sentences and changing to "…and calibration in Sec. 3-6. Similar procedures were applied in East and Southwest Chine but, for the sake of conciseness, are not shown in this study."
- Line 102: please, check if the right use of English is "over North China" instead of "in North China)
- Lines 102-103: could you indicate the area, in km², of these three regions?
- Line 104: consider changing to "the NCEP…dataset, which has…"
- Line 105 and 107: the use of "to the present" does not clarify the reader on what data period was actually used. Please, consider specifying the period of data used in the study.
- Lines 104-105: consider changing to "…and extending from…, was used for SST anomaly analysis.
- Line 108: consider changing to "…was used to verify the operational performance…"
- About CFSv2: I am missing a description in more details of the data used: forecasts issued when? Daily? At each month? For which maximum lead time? What did you use in the study: anomalies with respect to model climatology or precipitation forecasts? If anomalies, what was the climatology? Did you use an ensemble? If yes, how many ensemble members were used?
- Line 110: I suggest using "forecast" instead of "forecasted". Although both are correct, "forecast" is more common (even referring to the past).
- Line 111 (page 5): consider changing to "… forecast datasets were transformed into daily forecasts with a simple time-weighted mean". Also, what do you mean by "time weighted mean"? Wasn't it enough to add four blocks of 6-hour precipitation to have the 24-hour precipitation? What is "averaged" here?
- Fig. 2: consider indicating the North in the map and the relative position of the area in the world map. Consider deleting the word "study" when referring to the nine drought regions, as, for the reader, study regions are only the three regions of the study. Also, the light grey curves are not visible; consider not presenting them if they are not important for the comprehension of the study.
- Lines 118-125: this paragraph is very confusing to me. I suggest some re-writing as follows:
  - First, line 118: consider splitting the sentence into two: "in this study. The calculation period is…"
  - Line 119: then, consider adding here the last sentence. "… period is 1979-2014. The daily area-averaged precipitation datasets were first computed over the three study regions."

- Then add the way you computed the SPI3 from these daily values: "Usually, SPI3 values vary in a monthly time scale, i.e., each month a new value is determined from the precipitation totals of the previous three months …"
- Then, lines 120 onwards, I suggest changing to the following: "In this study, we chose to update the SPI3 daily, which was also recommended by…, i.e., every day a new value is determined from the precipitation totals of the previous three months. Specific illustration and details for …. are shown in Fig. 3." I suggest to end the paragraph here.

- Line 135: consider deleting the part "(e.g., severely dry)" as this is not needed for comprehension.
- Line 136: consider changing to "…to the slightly dry grade…"
- Line 144: why only North China? What about the other studied regions? Could you add the events for the other regions in Table 2? Or at least mention in the text the number of events identified to each grade?
- Line 152: consider changing to "…spells in order to further…
- Line 153-154: I do not fully understand this sentence. Are you referring tie the case in Fig. 4? Or is it valid for all cases due to the methodology adopted? Please, clarify. Also what do you mean by "special"?
- Line 177: isn't this a repetition of what is already said in line 156? Please, check if it is necessary or if you can shorten the text here.
- Line 191: consider changing to "…variables, which, on this study, are…"
- Line 196: consider changing to "…at each grid point".
- Line 198: consider changing to "…to decompose spatial-temporal datasets of…"
- Line 200: consider changing to "Considering that …, we focus on them in this study."
- Line 201: consider changing to "In addition, in order to…"
- Line 206: do you mean "Relevant results are shown in Fig. 7,…"? Considering changing "found" to "shown" if this is the case.
- Line 218: consider deleting the repetition of the word "area" and changing to "… positive pattern (Region B)…"
- Line 226: I do not understand what you mean by "ignored". Do you mean ignored in the analysis (i.e., not considered) or do you mean ignored in the illustrations (i.e., not shown here, but considered in the analysis). Consider changing to "…, the specified illustrations are not shown here" or "….were ignored in the analysis", according to the meaning you want to give here.
- Line 229: consider changing to "…used for the construction of the predictors".
- Line 240: consider changing to "In this study, it is used to build the synchronous…"
- Line 241: consider changing to "…and the prediction target SPI3."
- Line 245: I do not fully understand the first sentence. Do you mean the model is calibrated for every year, i.e., every year has a set of parameters? Please, clarify and check the use of the English language. The explanation in line 251 seems clearer.
- Line 247: correct to "Table 6" (not Table S1)
- Line 251: not fully clear to me. Consider changing to "…therefore, the number of samples used for calibration also increases year by year." Is that what you mean?
- Line 255: what do you mean by "key turning points and trends"? Consider either deleting this part or explaining it.

- Line 264, 270: consider changing to "forecast" instead of "forecasted", since it is more commonly used. Check for other uses in the text too.
- Lines 265-266: since the references for the datasets were already specified before, I think you do not need to add then here. They can be deleted.
- Line 266: this part can also be deleted: "… which is a type of climate forecast model", since you have already introduced the dataset in Sec. 2.
- Line 325: why do you use the word "eventually"? I would suggest deleting it.
- Line 438-439: consider changing to "…helped improve the clarity of the manuscript and made us think about …".
- Overall suggestion for Section Discussions or Conclusions: since the paper will be published on a special issue on seasonal hydrological forecasting, could you add a sentence on how the work presented on meteorological drought could be used for hydrological seasonal forecasting/outlook? Thanks!

Thank you for your manuscript!

Best regards,

Maria-Helena Ramos

*Guest Editor for the Special Issue: Sub-seasonal to seasonal hydrological forecasting*